# Estimating the snow depth, the snow-ice interface temperature, and the effective temperature of Arctic sea ice using Advanced Microwave Scanning Radiometer 2 and Ice Mass balance Buoys data

Lise Kilic[1], Rasmus Tage Tonboe[2], Catherine Prigent[1], and Georg Heygster[3]

[1]Sorbonne Université, Observatoire de Paris, Université PSL, CNRS, LERMA, Paris, France
[2]Danish Meteorological Institute, Copenhagen, Denmark
[3]Institute of Environmental Physics, University of Bremen, Bremen, Germany

**Correspondence:** Lise Kilic (lise.kilic@obspm.fr)

**Abstract.** Mapping Sea Ice Concentration (SIC) and understanding sea ice properties and variability is important especially today with the recent Arctic sea ice decline. Moreover, accurate estimation of the sea ice effective temperature ($T_{eff}$) at 50 GHz is needed for atmospheric sounding applications over sea ice and for noise reduction in SIC estimates. At low microwave frequencies, the sensitivity to atmosphere is low, and it is possible to derive sea ice parameters due to the penetration of microwaves in the snow and ice layers. In this study, we propose simple algorithms to derive the snow depth, the snow-ice interface temperature ($T_{Snow-Ice}$) and the $T_{eff}$ of Arctic sea ice from microwave brightness temperatures (TBs). This is achieved using the Round Robin Data Package of the ESA sea ice CCI project, which contains TBs from the Advanced Microwave Scanning Radiometer 2 (AMSR2) collocated with measurements from Ice Mass balance Buoys (IMBs) and the NASA Operation Ice Bridge (OIB) airborne campaigns over the Arctic sea ice. The snow depth over sea ice is estimated with an error of 5.1 cm using a multilinear regression with the TBs at 6V, 18V, and 36V. The $T_{Snow-Ice}$ is retrieved using a linear regression as a function of the snow depth and the TBs at 10V or 6V. The Root Mean Square Errors (RMSEs) obtained are 2.87 and 2.90 K respectively, with the 10V and 6V TBs. The $T_{eff}$ at microwave frequencies between 6 and 89 GHz is expressed as a function of $T_{Snow-Ice}$ using data from a thermodynamical model combined with the Microwave Emission Model of Layered Snow-packs. $T_{eff}$ is estimated from the $T_{Snow-Ice}$ with a RMSE of less than 1 K.

## 1 Introduction

In situ observations of the variables controling the sea ice energy and momentum balance in polar regions are scarce. One way to overcome this observational gap is to use satellites for measuring sea ice properties. The objective of this study is to estimate key sea ice variables from satellite remote sensing to improve current sea ice models and prediction, Sea Ice Concentration (SIC) mapping in the EUMETSAT Ocean and Sea Ice Satellite Application Facility (OSISAF) project, and polar atmospheric sounding applications.

Sea ice thermodynamics is controlled by the regional heat budget (Maykut and Untersteiner, 1971). In general, sea ice is covered by snow, which can reach a mean thickness of up to ~50 cm in the Arctic (Sato and Inoue, 2018). Snow on sea ice strongly affects the sea ice energy and radiation balance, with its high insulation of heat and reflectivity of solar radiation. Snow is a poor conductor of heat: it insulates the sea-ice and reduces the winter ice growth (Fichefet and Maqueda, 1999). In summer, its high albedo reduces the sea-ice melting rate. The high albedo of snow on sea ice compared to open water albedo plays an important role in the sea ice albedo feedback mechanism and Arctic amplification (Hall, 2004). Sato and Inoue (2018) suggest that the recent sea ice growth has been effectively limited by the increase in snow depth on thin ice during winter. Current sea ice models include snow schemes (e.g., Lecomte et al. (2011)), with the snow depth and temperature gradient in the snow pack modulating the sea ice growth and melt. Improved estimates of Snow Depth (SD), as well as Snow-Ice interface Temperature ($T_{Snow-Ice}$) from satellite observations would provide valuable information on the vertical thermodynamics in the snow and ice, to improve current sea ice models and therefore the prediction of sea ice growth.

Here we propose a simple algorithm to retrieve SD and $T_{Snow-Ice}$ from passive microwave observations from the Advanced Microwave Scanning Radiometer 2 (AMSR2), based on a large dataset of collocated *in situ* and satellite observations. An extensive Round Robin Data Package (RRDP) (Pedersen et al. (2018),https://figshare.com/articles/Reference_dataset_for_sea_ice_concentration/6626549) has been developed during the European Space Agency (ESA) sea ice Climate Change Initiative (CCI) project and the SPICES (Space-borne observations for detecting and forecasting sea ice cover extremes) project (http://www.seaice.dk/ecv2/rrdb-v1.1/). It contains *in situ* data from the Ice Mass balance Buoys (IMBs), and the Operation Ice Bridge (OIB) airborne campaigns collocated with AMSR2 brightness temperature measurements between 6 and 89 GHz.

Algorithms already exist to retrieve the snow depth from microwave observations. Markus and Cavalieri (1998) and Comiso et al. (2003) use the spectral gradient ratio of the 19 and 37 GHz (GR37/19) in vertical polarization to deduce the snow depth over sea ice. This method has been developed for dry snow on First Year Ice (FYI) in Antarctica, and it is applicable only to this ice type. Sea ice emissivity depends on the ice type. At frequencies $\geq$ 18 GHz, the ice emissivity is higher for FYI than for Multi Year Ice (MYI) (Comiso, 1983; Spreen et al., 2008). The difference of emissivity between the 19 and 37 GHz can be used to retrieve the snow depth or the sea ice type. Therefore, the snow depth algorithms which use this gradient ratio (GR37/19) are strongly dependent to the ice type. Improvements of Markus and Cavalieri (1998) have been suggested by Markus et al. (2011) and Kern and Ozsoy-Çiçek (2016). More recently, Rostosky et al. (2018) revisit the methodology for the Arctic region, using a new gradient ratio between 7 and 19 GHz (GR19/7), to derive snow depth over both FYI and MYI. For their study, they use the snow depth of OIB campaigns obtained in March and April. With the help of the RRDP, we will extend the methodology to the full winter (from December $1^{st}$ to April $1^{st}$) for the Arctic region using the IMB snow depth data.

Tonboe et al. (2011) showed from radiative transfer simulations that there is a high linear correlation between the $T_{Snow-Ice}$ and the passive microwave observations at 6 GHz. Preliminary results from Grönfeldt (2015) evidenced the possibility to derive the temperature of sea ice from passive microwave observations using simple regression models. This work will be extended here to estimate $T_{Snow-Ice}$ over Arctic sea ice.

Passive microwave satellite observations between 50 and 60 GHz are extensively used to provide the atmospheric temperature profiles in Numerical Weather Prediction (NWP) centers, with instruments such as the Advanced Microwave Sounding

Unit-A (AMSU-A) or the Advanced Technology Microwave Sounder (ATMS). For an accurate estimation of the temperature profile in the lower atmosphere, quantifying the surface contribution is required. The surface contribution i.e., the surface brightness temperature (TB) depends on the frequency, and it is the product of a surface effective emissivity ($e_{eff}$) and a surface effective Temperature ($T_{eff}$):

$$TB = e_{eff} \cdot T_{eff} \tag{1}$$

$T_{eff}$ is defined as the integrated temperature over a layer corresponding to the penetration depth at the given frequency: the larger the wavelength, the deeper the penetration into the medium. In the same way, $e_{eff}$ represents the integrated emissivity over a layer corresponding to the penetration depth. It depends on the frequency, the incidence angle, and the sub-surface extinction and reflections between snow and sea ice layers (Tonboe, 2010). Therefore, estimating the surface contribution is particularly complicated over sea ice, due to the layering and the vertical structure of the snowpack wich are affecting the microwave emission processes (Mathew et al., 2008; Rosenkranz and Mätzler, 2008; Harlow, 2009, 2011; Tonboe, 2010; Tonboe et al., 2011), and to the large spatial and temporal variability of sea ice and snow cover (English, 2008; Tonboe et al., 2013; Wang et al., 2017). The understanding of the relationship between $T_{eff}$ and the physical temperature profile is complicated, especially at microwave frequencies $\geq$18 GHz when scattering occurs, but it has been shown that from 6 to 50 GHz there is a high correlation between the $T_{eff}$ and the $T_{Snow-Ice}$ (Tonboe et al., 2011). With $T_{Snow-Ice}$ estimated from the AMSR2 observations, we will deduce the sea ice $T_{eff}$ at AMSR2 frequencies between 6 and 89 GHz, using linear regression.

Section 2 describes the dataset and the methodology used in this study. The snow depth retrieval is presented in Section 3. Section 4 reports on the $T_{Snow-Ice}$ retrieval. Finally, microwave sea ice $T_{eff}$ at 50 GHz is derived, for application to temperature atmospheric sounding (Section 5). Section 6 discusses the snow depth and the $T_{Snow-Ice}$ retrieval results over a winter in Arctic. Section 7 concludes this study.

## 2 Material and Methods

### 2.1 The database of collocated satellite observations and in situ measurements

The RRDP from the ESA sea ice CCI project is a dataset openly available (Pedersen et al. (2018), https://figshare.com/articles/ Reference_dataset_for_sea_ice_concentration/6626549). It contains an extensive collection of collocated satellite microwave radiometer data with *in situ* buoy or airborne campaign measurements and other geophysical parameters, with relevance for computing and understanding the variability of the microwave observations over sea ice. It covers areas with 0% and 100% of SIC and different sea ice types (thin ice, first-year ice, multiyear ice), for all seasons including summer melt. In our study, we will focus on Arctic sea ice during winter in regions with 100% sea ice cover. Two different datasets from the RRDP are used: AMSR2 brightness temperatures (TBs) collocated with IMB measurements, and AMSR2 TBs collocated with OIB airborne campaign measurements.

**Table 1.** List of the IMBs used in this study, with the mean snow depth (column 5) and the mean ice thickness (column 6) computed over the duration of the measurements (column 2).

| Buoy ID | Duration of measurements during winter (dd/mm/yy) | Deployment location | Position on December $1^{st}$ (lat; lon) | Mean snow depth (cm) | Mean ice thickness (cm) |
|---|---|---|---|---|---|
| 2012G | 01/12/12 - 06/02/13 | Central Arctic | (85,79°; -134,88°) | 34.1 | 162.8 |
| 2012H | 01/12/12 - 06/02/13 | Beaufort Sea | (80,39°; -129,23°) | 23.2 | 173.3 |
| 2012J | 01/12/12 - 06/02/13 | Laptev Sea | (82,87°; 139,09°) | 25.5 | 100.3 |
| 2012L | 01/12/12 - 06/02/13 | Beaufort Sea | (80,36°; -138,55°) | 8.5 | 330.1 |
| 2013F | 01/12/13 - 31/03/14 | Beaufort Sea | (76,15°; -146,27°) | 50.3 | 145.7 |
| 2013G | 01/12/13 - 31/03/14 | Beaufort Sea | (75,84°; -151,46°) | 21.3 | 249.4 |
| 2014F | 01/12/14 - 11/03/15 | Beaufort Sea | (76,32°; -143,10°) | 16.1 | 151.8 |
| 2014I | 01/12/14 - 12/03/15 | Beaufort Sea | (78,52°; -148,70°) | 22.6 | 155.3 |

AMSR2 is a passive microwave radiometer on board the JAXA GCOM-W1 satellite (launched in May 18, 2012). AMSR2 has 14 channels at 6.9, 7.3, 10.65, 18.7, 23.8, 36.5 and 89 GHz for both vertical and horizontal polarizations and it observes at 55° of incidence angle. In the RRDP, the spatial resolution of each channel is resampled by JAXA to the 6.9 GHz resolution (32×62 km ) (see AMSR2 L1R products, Maeda et al. (2011) and Maeda et al. (2016)) before collocation with buoy or airborne

campaign measurements (RRDP report, Pedersen and Saldo (2016) and Pedersen et al. (2018)).

IMBs are installed by the Cold Regions Research and Engineering Laboratory (CRREL) to measure the ice mass balance of the Arctic sea ice cover (Richter-Menge et al., 2006; Perovich and Richter-Menge, 2006). Buoy components include acoustic sounders and a string of thermistors. The thermistor string is extending from the air, through the snow cover and sea ice, into the water with the temperature sensors located every 10 cm along the string. It measures the physical temperature with an accuracy

of 0.1 K. There are two acoustic sounders located above the snow surface and below the sea ice. The acoustic sounders measure the position of snow and ice surfaces (top and bottom) with a precision of 5 mm, from which the snow depth is computed. The buoys also include instruments to measure air temperature, barometric air pressure and GPS geographical position (Perovich et al., 2017). Several IMBs are deployed by the CRREL at different locations and times during the year. We only use Arctic buoy data recorded during winter (December $1^{st}$ to April $1^{st}$) to avoid cases where ice starts to melt. The IMB available for

this study are all located on MYI, with an ice thickness $\geq$ 1 meter. A summary of buoy information corresponding to these criteria is given in Table 1 and the IMB locations are shown in Figure 1. IMB measurements collocated with AMSR2 TBs used in this study totalize 2845 observations.

For snow depth retrieval, we also used data from the OIB airborne campaign. The NASA OIB project has collected ice and snow depth data in the Arctic during annual flight campaigns (March-May) since 2009. The data are especially valuable in this

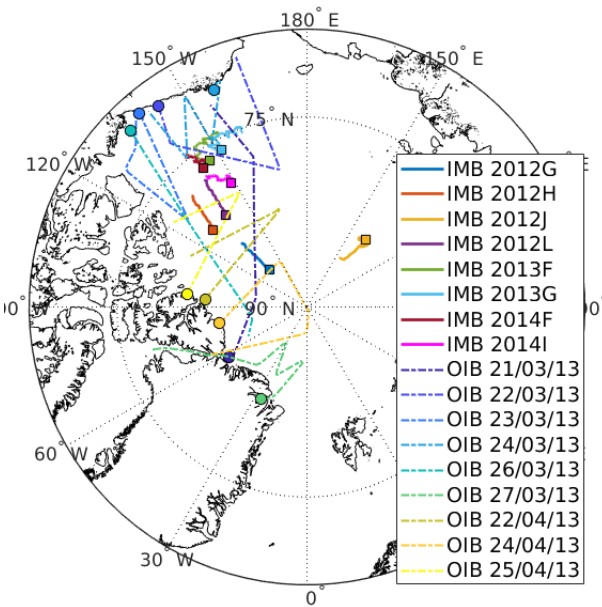

**Figure 1.** Ice Mass balance Buoy (IMB) and Operation Ice Bridge (OIB) flight locations over Arctic sea ice. Squares indicate the position of IMBs on December $1^{st}$ and circles indicate the starting points of OIB campaigns.

context since they contain snow depth information from the snow radar onboard the aircraft, not only from single points, but continuously along the flight path. The vertical resolution of the OIB snow radar is 3 cm, and the uncertainty on the snow depth is around 6 cm compared with in situ measurements (Kurtz et al., 2013). Recent studies evidence larger errors on OIB snow depth (Kwok and Maksym, 2014) with issues to detect snow depth under 8 cm (Kwok and Maksym, 2014; Holt et al., 2015).

These different limitations are summarized in Kwok et al. (2017). In the RRDP, the snow depth data from OIB snow radar are averaged into 50 km sections to be collocated with AMSR2 observations. For our study we use the OIB data from the 2013 campaign. It totalizes 408 observations over 8 days in March and April and covers FYI and MYI areas. Figure 1 summarizes the location of IMBs and OIB campaigns over the Arctic ocean.

It is important to note that there are discrepancies due to the scale, when comparing point measurements from buoys with

10 the spatially averaged data from satellites or aircrafts (Dybkjær et al., 2012).

## 2.2 The database of simulated effective temperature and brightness temperature from sea ice properties

For the estimation of $T_{eff}$, we use a microwave emission model coupled with a thermodynamic model. The emission model uses the temperature, density, snow crystal and brine inclusion size, salinity, and snow or ice type to estimate the microwave emissivity, the $T_{eff}$, and the TB of sea ice. It is coupled with a thermodynamic model in order to provide realistic microphysical

inputs. The thermodynamic model for snow and sea ice is forced with ECMWF ERA40 meteorological data input: surface air pressure, 2m air temperature, wind speed, incoming shortwave and longwave radiation, relative humidity, and accumulated

precipitation. It computes a centimeter scale profile of the parameters used as inputs to the emission model. The emission model used here is a sea ice version of the Microwave Emission Model of Layered Snowpacks (MEMLS) (Wiesmann and Mätzler, 1999) described in Mätzler (2006). The simulations were part of an earlier version of the RRDP and the simulation methodology is described in Tonboe (2010). This MEMLS simulation uses among its inputs the snow depth and the $T_{Snow-Ice}$
and compute $T_{effs}$ and TBs at different frequencies (from 1.4 to 183 GHz). The dataset contains 1100 cases and is called the MEMLS simulated dataset in the following.

## 2.3 Methodology

In this study, we propose simple algorithms, using multilinear regressions, to derive the snow depth, the $T_{Snow-Ice}$, and the $T_{eff}$ of sea ice from AMSR2 TBs.

The measurements from the IMB 2012G, 2012H, 2012J, and 2012L, collocated with AMSR2 TBs, are used as the training dataset for the different regressions to retrieve snow depth and $T_{Snow-Ice}$. These buoys have been selected because they are located in different regions across the Arctic and show a large range of snow depths. The measurements from IMB 2013F, 2013G, 2014F and 2014I which are all located in the Beaufort sea are used as the testing dataset.

First, the IMB snow depth is expressed as a function of the AMSR2 TBs using a multilinear regression (see Section 3.1).
The OIB data are used for the forward selection and the IMB training dataset is used to perform the regression. Second, the $T_{Snow-Ice}$ is expressed as a function of TBs and snow depth, using linear regressions. An automated method to detect the position of the snow-ice interface on the vertical temperature profile measured by the IMB thermistor string is developed (see Section 4.1). Then, the IMB training dataset is used to perform the regressions (see Section 4.3). For this part there are two consecutive regressions: the first one is done between the centered (the average was subtracted) $T_{Snow-Ice}$ and TBs ; the
second one is done between the $T_{Snow-Ice}$ corrected for the TB dependence and the snow depth. Third, the sea ice $T_{eff}$ at different microwave frequencies is expressed as a function of the $T_{Snow-Ice}$ (see Section 5.2). This final step is using the simulations from a thermodynamical model and MEMLS to derive linear regression equations for the $T_{eff}$ at frequencies between 6 and 89 GHz. The $T_{eff}$ at 50 GHz is of special interest for atmospheric sounding applications.

## 3 Snow depth estimation

## 3.1 Multilinear regression to retrieve the snow depth

A forward selection method is used to choose the best AMSR2 channels to retrieve snow depth. It is a statistical method to determine the best predictor combinations (here, AMSR2 TBs) to retrieve a variable (here, snow depth). We use the stepwise regression (Draper and Smith, 1998). It is a sequential predictor selection technique: at each step statistic tests are computed, and the predictors included in the model are adjusted. Our training dataset for this forward selection is the OIB snow depth
from the 2013 campaign included in the RRDP. OIB data are chosen for forward selection because the data cover a large area with a wide range of snow depths. In addition, the scale of the averaged OIB data is closer to satellite footprint than buoy

measurements, increasing the consistency with the satellite observations. Forward selection tests have also been done with the IMB training dataset but the results were not satisfactory. We find that the best channel combination for snow depth retrieval is the combination of the 3 channels at 6.9, 18.7, and 36.5 GHz in vertical polarization (6V, 18V, and 36V).

Then, a multilinear regression is conducted using the IMB training dataset (buoys G, H, J, L in 2012 collocated with AMSR2 TBs). The snow depth is given as a linear combination of the TBs at 6V, 18V, and 36V :

$$SD = 1.7701 + 0.0175 \cdot TB_{6V} - 0.0280 \cdot TB_{18V} + 0.0041 \cdot TB_{36V}, \tag{2}$$

with SD the snow depth expressed in m and TB in K. This model was trained with snow depths between 5 and 40 cm.

The forward selection has also been tested constraining the number of predictors to 2 and 4. The combinations obtained are: 18V and 36V for 2 channels, and 6V, 18V, 36V, and 89V for 4 channels. Then, the multilinear regression has been performed using these combinations of 2 or 4 channels. The results show that the 3 channel combination is the best in terms of RMSE and correlation compared to the 2 or 4 channel combination (see Section 3.2).

## 3.2 Results of the snow depth retrieval

Figure 2 shows the comparison between the observed snow depth measured by the acoustic sounder of IMB and the regressed snow depth computed from AMSR2 TBs with Eq. 2. The RMSE between the IMB snow depth observations and our snow depth regression is 12.0 cm and the correlation coefficient is 0.66, using the IMBs 2013F, 2013G, 2014F and 2014I (which are not in the training dataset). The buoy 2013F observes a large snow depth ($> 40$ cm) which is outside the bounds of our snow depth model. Tests are conducted to improve the estimation, including the 2013F buoy in the training dataset, with equal numbers of observations for different ranges of snow depths: it does not improve the results. Our model obtained the same snow depth estimation between buoys 2013G and 2013F. It is consistent because these buoys are spatialy very close. Therefore, we suspect that the 2013F buoy is located nearby a ridge or hummock where the local snow depth is large but not detectable at the satellite footprint scale. Without including the buoy 2013F in the computation, the RMSE for our snow depth model is 5.1 cm and the correlation coefficient is 0.61.

We also compare the snow depth retrievals with the measurements of the 2013 OIB campaigns (see Figure 3) with the ice type computed from the gradient ratio between 19 and 37 GHz (Baordo and Geer, 2015). Our snow depth regression (Eq. 2) RMSE is 6.26 cm and the correlation coefficient with OIB observations is 0.87. Note that the uncertainties on OIB data for the 2013 campaigns are between 2 cm and 22 cm with a mean Standard Deviation (StD) of 11 cm (OIB snow depth StD provided in the RRDP). Looking at Figure 3, our snow depth regression is applicable to both ice types. The RMSEs computed for MYI and FYI are respectively 7.2 cm and 3.9 cm, and the correlations are 0.71 and 0.03. The RMSE is smaller for FYI because the snow depth variability of FYI is also smaller. The low correlation obtained for FYI can come from the limited number of observations and because the snow depth variability observed is within the signal noise.

Spatial scales are different when comparing satellite measurements or airborne campaign measurements with buoy measurements. Discrepancies can appear due to the spatial variability of the snow depth. It can explain that the correlation is higher when comparing snow depth estimated from AMSR2 TBs with the snow depth observed from OIB radar. It is also important

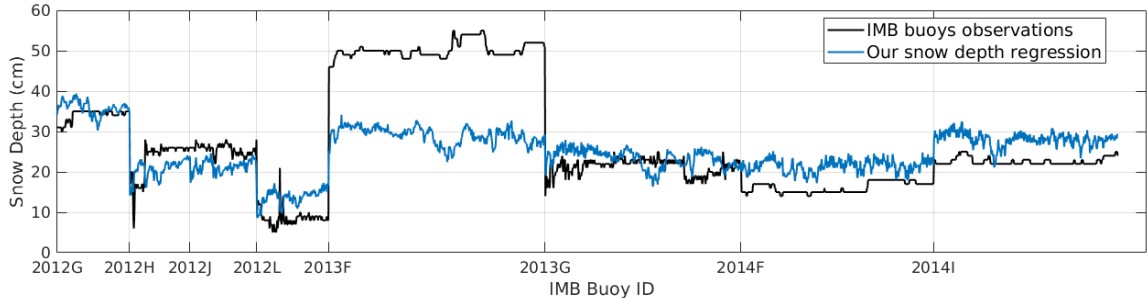

**Figure 2.** Time series of the comparison between snow depths from IMB observations and our multilinear regression (Eq. 2). The beginning of the measurements with a new IMB is indicated on the x-axis.

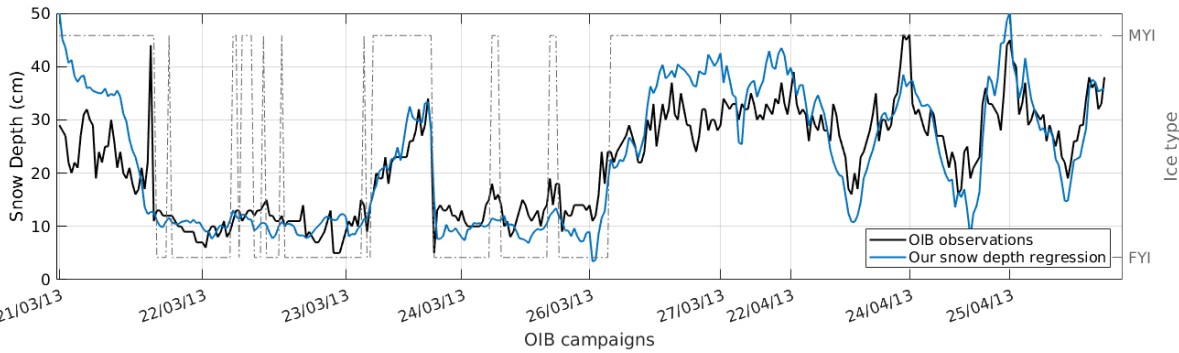

**Figure 3.** Time series of the comparison between snow depths (left y-axis) from OIB observations and our multilinear regression (Eq. 2). The beginning of the measurements with a new OIB campaign is indicated on the x-axis. For each measurement, the ice type is indicated in grey dashed line (right y-axis).

to note that the OIB campaign data are from late winter to beginning of spring (March to April), while IMB measurements are from winter (December to March). The snow depth regression being developed on IMB measurements, this small change in the season can contribute to the larger RMSE observed with OIB data.

## 4 Snow-ice interface temperature estimation

### 4.1 Automatic interface position detection

During winter, the air temperature is very cold meaning that the snow surface temperature is cold compared to ice and water temperatures. Through sea ice, the temperature profile is piecewise linear and temperature increases with depth (see Figure 4). In the air, the temperature gradient is small because of turbulent mixing. In the snow, the temperature gradient is larger due to the thermal properties of snow. Therefore, air-snow and snow-ice interface positions can be detected by changes in the

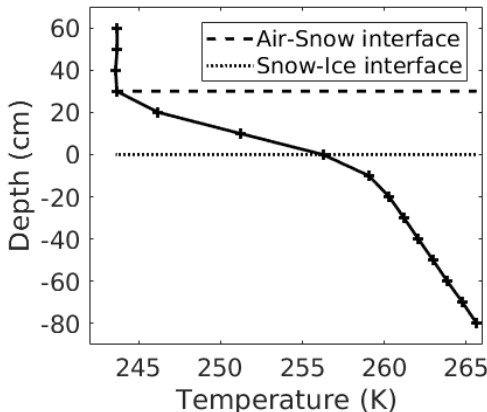

**Figure 4.** Averaged temperature profile (From December to February) measured by the IMB 2012G, with air-snow and snow-ice interface levels detected with our automated method.

temperature gradient. At the snow-air interface, the second derivative of the temperature profile reaches a maximum. At the snow-ice interface, the temperature gradient being lower in the ice than in the snow, the second derivative of the temperature profile reaches a minimum. Using these properties of the sea ice temperature profile, an automated method is implemented to detect the air-snow and the snow-ice interface positions in the temperature profile measured by the buoy thermistor string.

Figure 4 shows an averaged temperature profile through sea ice during winter, with the air-snow and snow-ice interface positions detected with our automated method. This method performs best during winter when the air is cold. It may not be applicable if the snow depth is lower than the vertical resolution of the thermistor string (10 cm), or if sea ice starts to melt and the temperature profile develops gradually toward an isothermal state. The method selects the thermistor which is located the closest to the interface. Note that the real interface position can be located between two thermistors. Therefore, the

shift between the real interface position and the thermistor the closest to the interface can be up to 5 cm. This can introduce uncertainties in our $T_{Snow-Ice}$ regression.

## 4.2    Correlation between the brightness temperature and the snow-ice interface temperature

During winter, the vertical position of the snow-ice interface is fixed with respect to the buoy thermistor string. The thermistor string is frozen into the ice which means that the thermistor at the snow-ice interface will stay at that interface unless there

is surface melt or snow ice formation and this rarely happens during winter. For each IMB, the snow-ice interface is detected with our automated method described in Section 4.1.

     We use a correlation analysis to select the TBs at different frequencies describing the variability of the $T_{Snow-Ice}$. Figure 5 shows the correlation coefficient between $T_{Snow-Ice}$ and AMSR2 TBs computed using the data from all IMBs (Table 1). The 89 GHz TBs are highly correlated with the air temperature (R>0.75). The 18.7, 23.8 and the 36.5 GHz TBs have a low

correlation with $T_{Snow-Ice}$ because of microwave scattering in the snow and/or shallow microwave penetration into the snow.

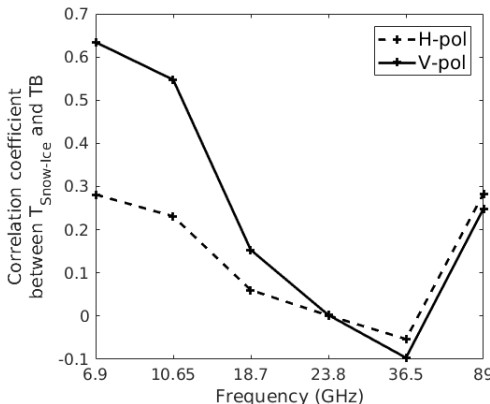

**Figure 5.** Correlation coefficient between the $T_{Snow-Ice}$ from IMBs and the AMSR2 TBs, as a funtion of AMSR2 frequency.

The 7.3 GHz channel is ignored because it contains practically the same information as the 6.9 GHz channel. The TBs at 6.9 and 10.65 GHz at vertical polarization, have the highest correlation with $T_{Snow-Ice}$ (R>0.5). Therefore the 10.65 and the 6.9 GHz at vertical polarization (10V and 6V) channels are selected as inputs to the linear regression to retrieve the $T_{Snow-Ice}$.

### 4.3 Linear regressions to retrieve the snow-ice interface temperature

To express the $T_{Snow-Ice}$ as a function of the TB at 6V and 10V, the linear regressions are calculated on centered data (i.e. the anomaly). For each buoy, the averaged $T_{Snow-Ice}$ is subtracted from the $T_{Snow-Ice}$ measurements and the same is done with the TB measurements. Thus, the temperature offset between the buoys is removed and the slope of the linear regression is unchanged:

$$\Delta T_{Snow-Ice} = a_1 \cdot \Delta TB_{6Vor10V} \Leftrightarrow T_{Snow-Ice} = a_1 \cdot TB_{6Vor10V} + offset_{buoy} \tag{3}$$

with $\Delta T_{Snow-Ice}$ and $\Delta TB$ describing the centered $T_{Snow-Ice}$ and TB. Figure 6 shows the linear regression between the $T_{Snow-Ice}$ and the TB at 6V and 10V, using the measurements from buoys 2012G, 2012H, 2012J, and 2012L. The slope coefficients ($a_1$) estimated between the $T_{Snow-Ice}$ and the TB at 6V and 10V are 1.086±0.020 and 1.078±0.019 respectively.

The offset ($offset_{buoy}$) in the linear regression equations between $T_{Snow-Ice}$ and the TB is different for each buoy, because it depends on the snow depth. The $T_{Snow-Ice}$ dependence on snow depth can be explained by the thermal insulation of snow

(Maaß et al., 2013; Untersteiner, 1986). Here, we establish an empirical relationship between the $T_{Snow-Ice}$ corrected of the TB linear dependence at 10V or 6V, and the snow depth as follows:

$$T_{Snow-Ice} - a_1 \cdot TB_{10V \ or \ 6V} = a_2 \cdot f(SD) + a_3, \tag{4}$$

with $f(SD)$ a function of snow depth.

Three different linear regressions have been tested to relate the $T_{Snow-Ice}$ using: the snow depth directly, the inverse of

the snow depth, and the logarithm of snow depth. Figure 7 shows the $T_{Snow-Ice}$ corrected from TB dependence as a function

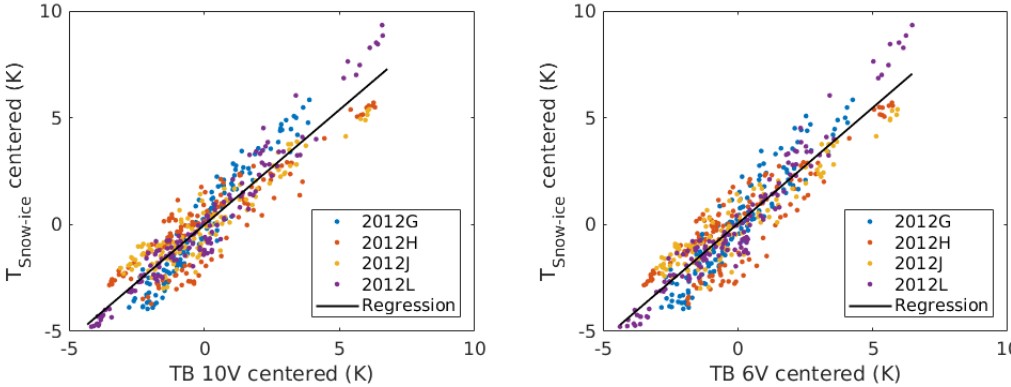

**Figure 6.** Centered $T_{Snow-Ice}$ expressed as a function of the centered TBs at 10V (left) and 6V (right). Data from the IMBs are in different colors depending on the buoy and the linear regression is the solid black line.

of snow depth. The different regressions are tested using the training dataset (IMB G, H, J, and L in 2012). The regression showing the best results uses the logarithm of the snow depth (solid black line in Fig. 7). The linear regression using the snow depth directly (red dashed line in Fig. 7) leads to an overestimation of the $T_{Snow-Ice}$ for large snow depth. The regression using the inverse of the snow depth (red dotted line in Fig. 7) leads to an underestimation for small snow depth. The RMSEs

obtained on the $T_{Snow-Ice}$ are compared and the relation using the logarithm of snow depth shows the lowest RMSE. Based on these results, the final equations to relate the $T_{Snow-Ice}$ to the snow depth and the TB at 10V and at 6V are:

$$T_{Snow-Ice} = 1.078 \cdot TB_{10V} + 5.67 \cdot log(SD) - 5.13 \tag{5}$$

$$T_{Snow-Ice} = 1.086 \cdot TB_{6V} + 3.98 \cdot log(SD) - 10.70 \tag{6}$$

where $T_{Snow-Ice}$ and TB are expressed in K, and SD is expressed in m.

### 4.4   Results of the snow-ice interface temperature retrieval

Figure 8 shows the comparisons between the observed $T_{Snow-Ice}$ and the regressed $T_{Snow-Ice}$ using the 10V and 6V TBs (Eq. 5 and 6), and the *in situ* snow depth measured by the acoustic sounder of IMB. The RMSEs are computed using the IMB 2013F, 2013G, 2014F, and 2014I. The regression of the $T_{Snow-Ice}$ using the *in situ* snow depth with the 10V TBs (Eq. 5) is

slightly better (RMSE = 1.78 K) than the regression with the 6V TBs (Eq. 6) (RMSE = 1.98 K). The variability due to the snow depth is better described with the regression using the 10V TBs. Figure 9 is the same as Figure 8 but with our snow depth estimation (Eq. 2). The RMSEs are 2.87 K for the 10V regression and 2.90 K for the 6V regression. The results are degraded because of the snow depth regression especially for the buoys with thick snow (~50 cm) or thin snow (~5 cm) (e.g., buoy 2013F and buoy 2012L). Note that the regression is tested with IMBs which are all located on MYI. However, our algorithm

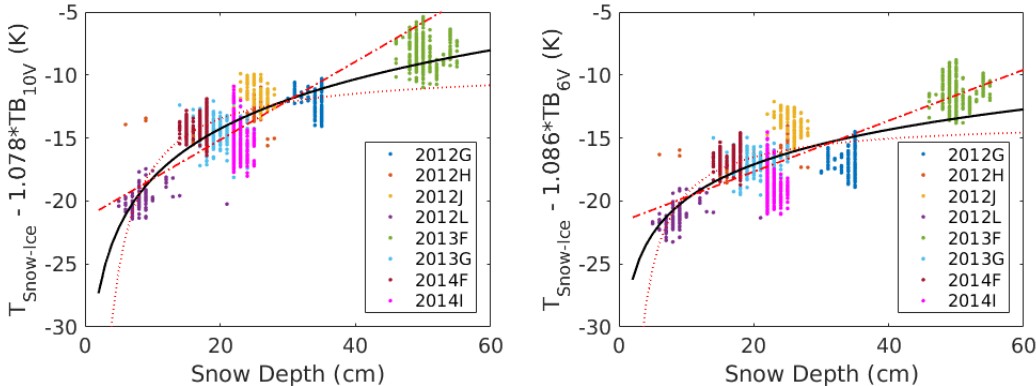

**Figure 7.** $T_{Snow-Ice}$ corrected of the 10V TB (left) and of the 6V TB (right) dependence as a function of snow depth. Data from the IMBs are represented by different colors, the regression using the snow depth is the dashed red line, the regression using the inverse of snow depth is the red dotted line, and the regression using the logarithm of the snow depth is the solid black line.

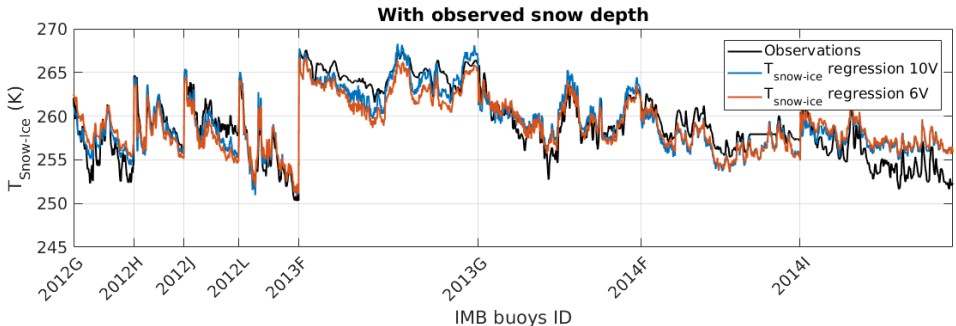

**Figure 8.** Time series of the comparisons between $T_{Snow-Ice}$ observations from IMBs (black line), and $T_{Snow-Ice}$ regressions with TBs at 10V (blue line) and at 6V (red line). The snow depth used in Eq. 5 and 6 is the snow depth observed by the IMB sounder. The beginning of the measurements with a new IMB is indicated on the x-axis.

to derive the $T_{Snow-Ice}$ is also applicable over FYI areas, as our snow depth algorithm is applicable to both ice types and our $T_{Snow-Ice}$ algorithm uses the channels 10V or 6V which have a limited sensitivity to the ice type (Comiso, 1983; Spreen et al., 2008).

## 5    Sea ice effective temperature estimation

### 5.1    Bias between the model and the observations

$T_{eff}$ is related to the frequency and the incidence angle of the satellite observations. It is not a geophysical variable that we can measure directly as an *in situ* parameter. A microwave emission model has to be used to computed the $T_{effs}$ from the

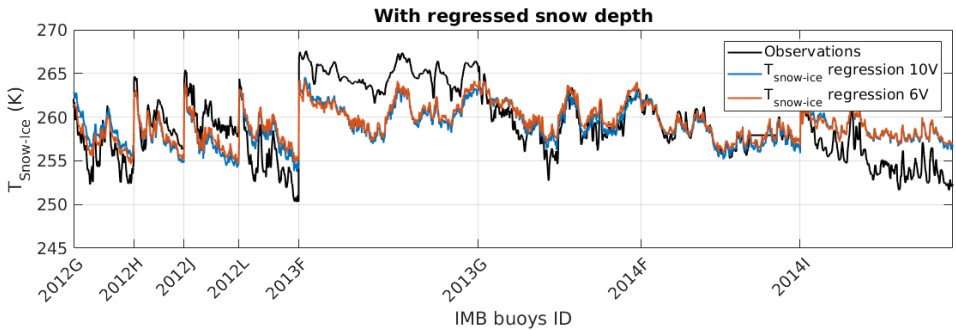

**Figure 9.** Same as Figure 8, using the regressed snow depth (Eq. 2) in place of in situ snow depth

geophysical parameters. The $T_{eff}$ used here is available from a simulated dataset using a thermodynamical model and the microwave emission model MEMLS. The model set-up and the simulations are described in Tonboe (2010). In this dataset, the TBs and the $T_{eff}s$ are simulated using the $T_{Snow-Ice}$ and the input snow and ice profiles from the thermodynamical model. Even though the simulated TB data are comparable to observations in terms of mean and standard deviation, both

the thermodynamical model and the emission model are based on physical equations and are not tuned to observations. TBs simulated with MEMLS are not fitted to AMSR2 TBs meaning that a bias is expected between the $T_{Snow-Ice}$ of the MEMLS simulated dataset ($T_{Snow-Ice\ MEMLS}$) and the $T_{Snow-Ice}$ estimated with our regression.

The bias obtained is the mean value of the difference between the $T_{Snow-Ice\ MEMLS}$, and the $T_{Snow-Ice}$ regressed from Eq. 5 and 6 using the TBs of the MEMLS simulated dataset as inputs. Biases of 3.97 K and 4.01 K are estimated, for the re-

gressions with 10V and 6V respectively. The RMSEs computed between the $T_{Snow-Ice\ MEMLS}$ and the $T_{Snow-Ice}$ regressed and corrected of the bias at 10V and 6V are 2.7 K and 2.07 K, respectively.

Figure 10 shows the $T_{Snow-Ice}$ from MEMLS simulated dataset as a function of TB at 10V and 6V, and the $T_{Snow-Ice}$ computed from our regressions (Eq. 5 and 6), with and without the bias correction. We can see that the slopes of our linear regressions are consistent with the data simulated from MEMLS.

**5.2   Linear regression between the effective temperature and the snow-ice interface temperature**

The $T_{eff}$ near 50 GHz in vertical polarization is correlated with the $T_{Snow-Ice}$ (Tonboe et al., 2011) and it can be expressed as a linear function of the $T_{Snow-Ice}$:

$$T_{eff(freq,pol)} = b_{1(freq,pol)} \cdot T_{Snow-Ice\ MEMLS} + b_{2(freq,pol)} \tag{7}$$

with $T_{eff}$, $b_1$ and $b_2$ depending on the frequency ($freq$) and on the polarization ($pol$). We use the MEMLS simulated dataset to

calculate the linear regression between the $T_{Snow-Ice}$ and the $T_{eff}$ at 6.9, 10.65, 18.7, 23.8, 36.5, 50, and 89 GHz in vertical polarization. $T_{eff}s$ at vertical and horizontal polarizations are about the same. Only the vertical polarization is considered here, because TBs measurements are noisier at horizontal polarization due to the variability of sea ice emissivity at this polarization.

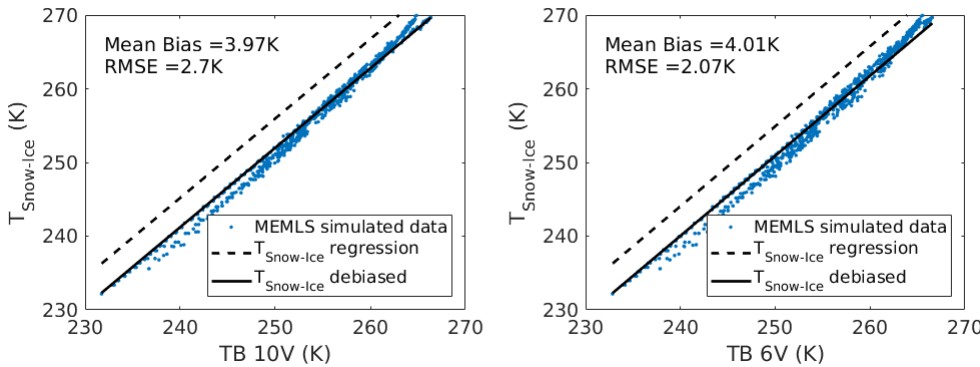

**Figure 10.** Comparisons between the $T_{Snow-Ice\ MEMLS}$ from the MEMLS simulated data in blue points, the regressed $T_{Snow-Ice}$ (Eq. 5 and 6) in dashed black line, and the regressed $T_{Snow-Ice}$ debiased to fit the MEMLS simulations in solid black line at 10V (left) and 6V (right) channels.

Figure 11 shows the $T_{eff}$ at 50V as a function of $T_{Snow-Ice}$. The linear regressions between the $T_{Snow-Ice}$ and the $T_{eff}$ at different frequencies are computed. The coefficients $b_1$ and $b_2$ of Eq. 7 are given in Table 2. The slope coefficient of the regression increases with frequency, meaning that the sensitivity of the $T_{eff}$ to the $T_{Snow-Ice}$ is increasing with frequency between 6 and 89 GHz. A slope coefficient lower than 1 means that the penetration depth at the given frequency is deeper than
snow-ice interface. At 50 GHz the slope coefficient is near to 1, meaning that the penetration depth is close to the depth of the snow-ice interface. The RMSEs are below 1 K, with the regression of $T_{eff}$ at 50V showing the lowest RMSE (0.33 K), and at 89V showing the highest RMSE (0.92 K).

These linear regressions between the $T_{eff}$ and the $T_{Snow-Ice\ MEMLS}$ (Eq. 7) are the final step to retrieve the $T_{eff}$ of sea ice at microwave frequencies as a function of TBs, using the work in the previous sections to express the $T_{Snow-Ice}$ as
a function of TBs (Eq. 2, and Eq. 5 or 6). The biases between the AMSR2 observations and the MEMLS simulated dataset are taken into account replacing $T_{Snow-Ice\ MEMLS}$ by $T_{Snow-Ice}$ estimated from AMSR2 TBs with a bias correction (see Table 2):

$$T_{eff(freq,pol)} = b_{1(freq,pol)} \cdot (T_{Snow-Ice} - 3.97) + b_{2(freq,pol)},\ \text{for the regression using 10V TB} \tag{8}$$

$$T_{eff(freq,pol)} = b_{1(freq,pol)} \cdot (T_{Snow-Ice} - 4.01) + b_{2(freq,pol)},\ \text{for the regression using 6V TB} \tag{9}$$

## 6   Discussion

For days in November, January, and April, Figure 12 shows the maps of the snow depth estimated with our multilinear regression (Eq. 2), the $T_{Snow-Ice}$ estimated with our multilinear regression (Eq. 5), and the MYI concentration products from

**Table 2.** Regressions of the $T_{eff}$ for different frequencies at vertical polarization as a function of the $T_{Snow-Ice}$ (see Eq. 7) using the MEMLS simulated dataset.

| Frequency (GHz) | slope coefficient $b_1$ | offset (K) $b_2$ | RMSE (K) |
|---|---|---|---|
| 6.9 | 0.888 | 30.2 | 0.89 |
| 10.7 | 0.901 | 26.6 | 0.75 |
| 18.7 | 0.920 | 21.5 | 0.63 |
| 23.8 | 0.932 | 18.4 | 0.57 |
| 36.5 | 0.960 | 10.9 | 0.41 |
| 50 | 0.989 | 2.96 | 0.33 |
| 89 | 1.06 | -16.4 | 0.92 |

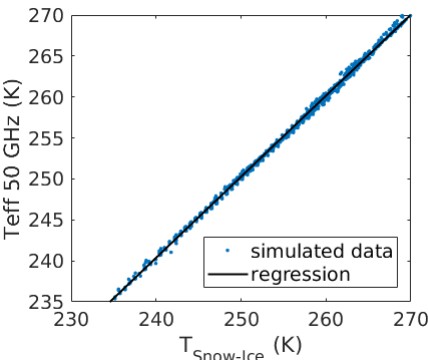

**Figure 11.** Regression of the $T_{eff}$ as a function of $T_{Snow-Ice}$ at 50 GHz in vertical polarization. The data from the MEMLS simulations are in blue points and the linear regression is the solid black line.

the University of Bremen (https://seaice.uni-bremen.de). Maps of the MYI concentration from University of Bremen are derived from AMSR2 and from the Advanced SCATterometer (ASCAT) with the method of Ye et al. (2016a, b). To perform our regressions, we use the AMSR2 TBs (Level L1R) provided by JAXA and the SIC from the European Centre for Medium-Range Weather Forecasts (ECMWF) Re-Analysis Interim (ERA-Interim) data. Only the areas with 100% SIC are considered
5    to compute the snow depth on sea ice and the $T_{Snow-Ice}$ with our method.

The results show that the snow depth is larger (40 cm) in the north of Greenland (Warren et al., 1999; Shalina and Sandven, 2018) due to the presence of drift snow caused by the numerous pressure ridges present in this area (Hanson, 1980), as

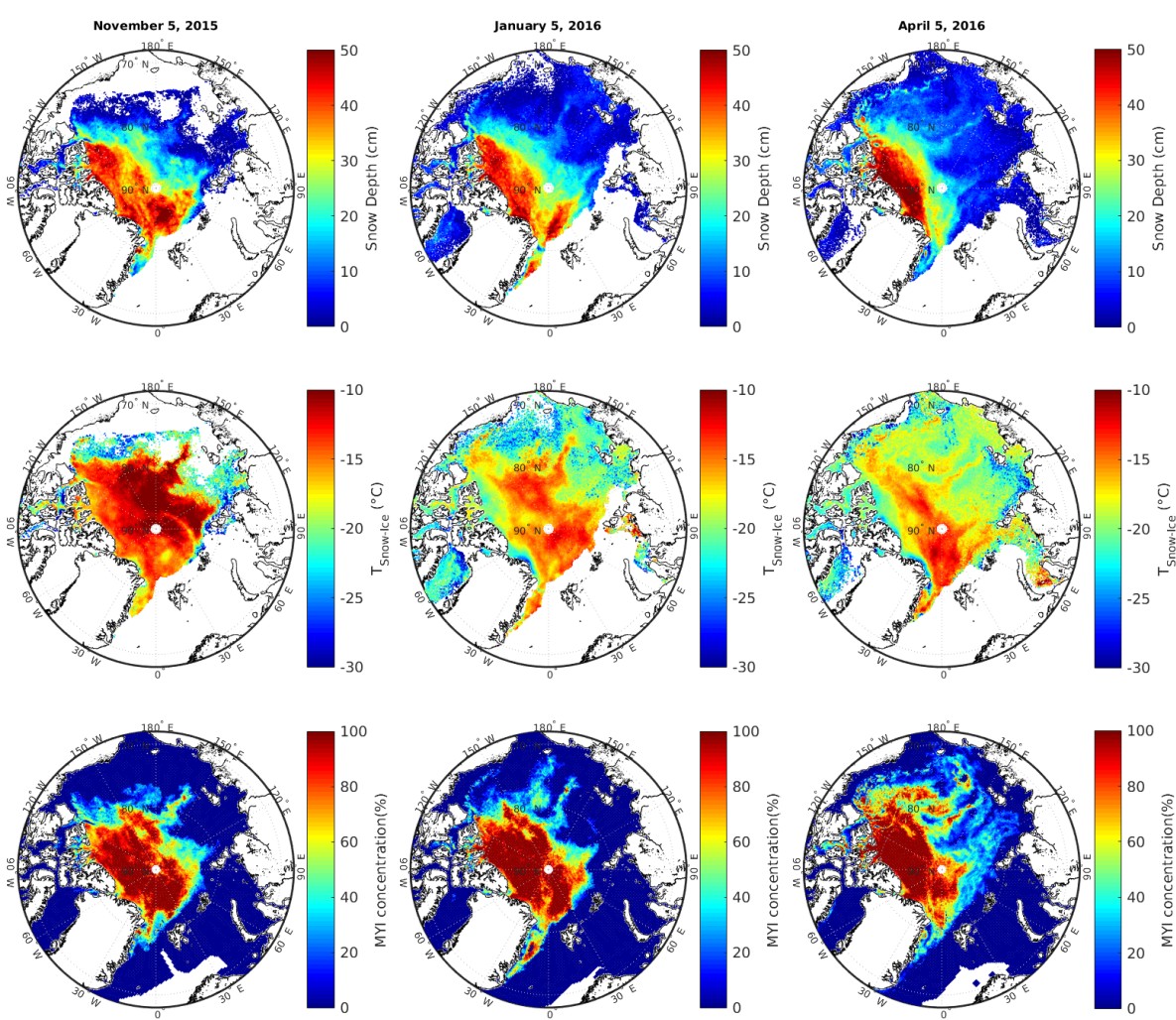

**Figure 12.** Maps of the snow depth (first row) and the $T_{Snow-Ice}$ (second row) estimated from our multilinear regression using AMSR2 TBs, with MultiYear Ice (MYI) concentration products (third row) from the University of Bremen on November 5, 2015 (left), January 5, 2016 (middle) and April 5, 2016 (right).

anticipated. We can observe that the snow depth is larger in areas with larger MYI concentrations. The variability of the snow cover is low during winter, as the snow depth reaches a maximum by December and remains relatively unchanged until snowmelt (Sturm et al., 2002).

For $T_{Snow-Ice}$, in January and April when the air temperature is cold (between -20 and -30°C over the whole Arctic, on
5   January 5 and April 5, 2016 from ERA-Interim air temperature), the areas with large snow depth show larger $T_{Snow-Ice}$
because of the thermal insulation power of the snow. It is different in November: the air temperature is warmer ($\sim$ -5°C near

Kara sea, $\sim$ -15°C near Laptev sea and $\sim$ -25°C in central Arctic and Beaufort sea, on November 5, 2015 from ERA-Interim air temperature) and the areas with thinner snow show larger $T_{Snow-Ice}$ which are close to the air temperature (Perovich and Elder, 2001). Note that we can observe low $T_{Snow-Ice}$ in some locations near the sea ice margins due to the presence of open ocean in the satellite footprint. As the brightness temperature of open water is low, the total brightness temperature measured is decreased and it impacts our $T_{Snow-Ice}$ estimation.

Visually the $T_{Snow-Ice}$ shows a high correlation with the distribution patterns of multiyear ice concentration of the same days: the highest values are found in the north of Greenland and in the Canada Basin, with some branches of higher values extending from there towards the Siberian coast, marking the Beaufort gyre of the Arctic sea ice drift (see the animations for the same year at https://seaice.uni-bremen.de/multiyear-ice-concentration/animations/). The main differences between FYI and MYI are, on average, the higher thickness of MYI and its higher snow load. Both effects will influence the $T_{Snow-Ice}$. Under the same conditions, a higher ice thickness will lead to a lower $T_{Snow-Ice}$. In contrast, it will be higher if only the snow depth is increased. The positive correlation between MYI concentration and $T_{Snow-Ice}$ suggests that the influence of the higher snow depth on MYI outbalances that of the higher ice thickness on the $T_{Snow-Ice}$, emphasizing the important role of snow on sea ice in its thermodynamic balance.

The similar patterns observed between the maps of the $T_{Snow-Ice}$ and the MYI concentration on Figure 12 are encouraging and gives confidence in the methodology developed here, as these MYI concentration products are from an independent work done at the University of Bremen and distributed daily to users. However it should be noted that the input channels of both methods overlap in some AMSR2 channels, and even different channels show some covariance (Scarlat et al., 2017).

## 7 Conclusions

We derive simple algorithms to estimate sea ice parameters such as the snow depth, the $T_{Snow-Ice}$, and the $T_{eff}$ of sea ice at microwave frequencies, from AMSR2 channels. This is achieved using the ESA RRDP which contains AMSR2 data collocated with IMB data and OIB campaign data. In addition, simulated TB outputs from a sea ice version of MEMLS are used for the regression of the $T_{eff}$. All the equations to retrieve these sea ice parameters are derived using several linear and multilinear regressions.

Our regression to retrieve the snow depth over winter Arctic sea ice uses the TBs at 6.9, 18.7 and 36.5 GHz in vertical polarization. A RMSE of 5.1 cm is obtained between the estimated and the IMB snow depths using an independent IMB test dataset. This snow depth retrieval is applicable to FYI and MYI, with lower uncertainties for FYI than for MYI (3.9 cm compared to 7.2 cm). To retrieve the $T_{Snow-Ice}$, two relations are derived using two different AMSR2 channels (10V or 6V) and the estimated snow depth. The two regressions show similar results. The errors are 2.87 K and 2.90 K respectively at 10V and 6V. This $T_{Snow-Ice}$ retrieval has been tested only for MYI. It can also be applied to FYI as the 6V and 10V channels have a limited sensitivity to the ice type (Comiso, 1983; Spreen et al., 2008). Finally the $T_{effs}$ at 6.9, 10.65, 18.7, 23.8, 36.5, 50, and 89 GHz in vertical polarization are retrieved as a function of $T_{Snow-Ice}$ using linear regressions. At the final step, the RMSEs of the linear regressions between the simulated $T_{Snow-Ice}$ and the $T_{eff}$ for all channels are lower than 1 K, with

a minimum value of 0.33 K at 50 GHz which is a key frequency for atmosphere temperature retrieval. The methodology to estimate snow depth and $T_{Snow-Ice}$ has been applied to several days during a winter season. It shows consistent results with MYI concentration estimates obtained independently.

These algorithms can be used to create snow depth and $T_{Snow-Ice}$ products which can improve the study of sea ice variability (e.g., sea ice growth). Informations on the $T_{Snow-Ice}$ may help in sea ice models by constraining the sea ice temperature gradient and the thermodynamical ice growth. The $T_{eff}$ estimations can be used in atmospheric radiative transfer calculations and to reduce noise in SIC retrieval algorithms (Tonboe et al., 2013) (e.g., EUMETSAT OSISAF global SIC product).

*Author contributions.* This study was conducted by L.K. and supervised by R.T.T. and C.P.. G.H. contributed to the analysis and to the correction of the draft.

*Competing interests.* The authors declare no conflict of interest.

*Acknowledgements.* This research was funded by EUMETSAT OSISAF (OSI VS17 03). The authors acknowledge the support from the EUMETSAT OSISAF visiting scientist program and the Danish Meteorological Institute for its welcome. We also acknowledge the reviewers for their precious comments which improved a lot this manuscript.

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
