# Peer review of "Estimating the snow depth, the snow-ice interface temperature, and the effective temperature of Arctic sea ice using Advanced Microwave Scanning Radiometer 2 and Ice Mass balance Buoys data"

_The Cryosphere, 2018_

## Referee Comment (RC1) · Pedersen (Referee) · 10 Dec 2018

Review of Lise Kilic et al. Estimating the snow depth, the snow-ice interface temperature, and the effective temperature of Arctic sea ice using Advanced Microwave Scanning Radiometer 2 and Ice Mass Balance buoys data.

General comments:

More discussion about the impact of ice type on the results should be included. The Markus & Cavalieri snow depth algorithm is only supposed to work properly over first

year ice, most of the OIB and IMB data are from areas of multi-year ice. These issues and their impact on the results should be more clearly identified and discussed.

There should be a clearer wording about when the results for Tsnow-ice are derived using in-situ snow depth and when they are derived using the estimated snow-depth from this study. Both in the abstract and in the conclusions, error numbers assuming in-situ snow depth measurements are given, but these are not generally available, so the uncertainties for the retrievals using satellite snow depths are generally more relevant.

The concept of effective temperature is based on an assumption of constant emissivity. It is here even referred to as surface emissivity. In reality the emissivity varies with depth as does the temperature, and in particular the emissivity at the surface is small since the emissivity of snow is very small during Winter (no absorption = no emissivity). It should be better explained what is actually the emissivity referred to as the surface emissivity, and some considerations about it's variability with temperature and salinity for example would be appreciated.

More detailed comments:

P1L20: Sea ice dynamics and thermodynamics -> Sea ice thermodynamics

P2L1: reduced -> reduces

P2L9: Advance -> Advanced

P2L11 and reference section: The RRDP should be referred to as Pedersen et al, 2018, https://figshare.com/articles/Reference_dataset_for_sea_ice_concentration/6626549

P2L24: In principle this should also be "surface effective emissivity" (see above), and it should be better explained how to estimate this emissivity.

P3L5: See comment P2L11 above

P3L10-11: Note that neither the OIB nor the IMB data in the RRDP are guaranteed 100% ice. This should be considered and the impact on the results should be discussed.

P3L15: See P2L11 above. In addition the resolution matching of AMSR2 is carried out by JAXA and should be referred to as Maeda et al, 2011 Maeda, K., Y. Taniguchi and K. Imaoka, (2016), GCOM-W1 AMSR2 Level 1R Product: Dataset of Brightness Temperature Modified Using the Antenna Pattern Matching Technique, IEEE Transactions on Geoscience and Remote Sensing, VOL. 54, NO. 2.

P3L19-20: The acoustic sounder only measures the position of the snow surface. The position of the ice surface is assumed from deployment or from the Summer measurements at the end of the ablation period. The sensor is mounted on a pole frozen into the ice, looking down at the snow surface. It measures distance between the instrument and the snow surface, thus recording the changes in the snow depth.

P3L21: IMB buoys -> IMBs. The B in IMB means Buoy and does not have to be repeated. There are many instances of this in the text.

P3L23: bouys -> buoy

P3L29: OIB radar -> the OIB snow radar. OIB operates other radars as well.

P5L1-5: Please include a bit more details about the simulated data, such as number of datapoints, types of ice etc.

P5L30: satisfying -> satisfactory

P6L1-10: Discuss also the potential for a seasonal variation in the regression. OIB data are all from late Winter to Spring, whereas the IMB data are for all Winter. What impact could that have, and why do you expect your regression from OIB to work also during other parts of the Winter.

P6L21-27: Here you need to discuss why you think the Markus and Cavalieri snow depth algorithm can be applied to MY-ice.

P6L31: Please provide a reference to the OIB uncertainties quoted here. Also note

that the RRDP OIB dataset contains information about the variability of the snow depth over the 50 km sections. This could have been used to filter out the OIB data with too much variability. The RRDP also contains ASCAT C-band scatterometer data that could be used to distinguish ice types.

Figure 2: You should not apply the Markus and Cavalieri algorithm to MY-ice and you should discuss the importance of ice type for your own snow depth retrievals.

P7L7-8: The temperature gradient is a function of the thermal conductivities and the depth of snow and ice respectively. The temperature gradient in snow is certainly not always 35 K/m! Please rephrase this sentence.

Section 4.1: This methodology is rather crude. It assumes thermodynamic equilibrium (which is not always the case, please discuss), and it could have been refined to a better estimate of the snow/ice interface temperature by the method outlined in section 4.1.5 of the RRDP manual (identifying the crossing point of the linear temperature profile in ice and in snow respectively). This might have reduced the quantization 'noise' in the IMB Tsnow-ice data.

P10L5-14: Equations (2), (3) and (4) do not make sense as they stand. The TBs should have been delta-TBs and you should specify the center TBs you subtracted to get to the delta TB and you should more clearly specify that these are NOT TBs.

P11L5-7: This should have been mentioned earlier and could have been fixed by applying the method from the RRDP manual described above under Section 4.1.

P12L3: Explain a bit more what Teff is and why you need simulated data.

P12L4: are simulated together -> are all simulated

P12L10: simulated data -> simulated TB data

P12L13-15: These biases are presumably in the MEMLS simulations and not in the TB data, so you should bias-correct the MEMLS simulations and not the AMSR2 TB data.

(This applies to figure 10)

P12L20-21: Explain more (f.ex using a reference) why H pol TBs are more noisy?? Figure 11: The figure text must be wrong. This figure must be for only one frequency (which)?

P13L4+5: As stated in the general comments, all layers emit, to the concept of "an" emitting layer is an abstraction and should be explained more carefully.

P13L8: section -> sections

P14L10-11: According to Warren (1999) the snow depth in general is not supposed to decrease from November to January, so this reference seems wrong. If this behavior is seen in certain regions please be more specific.

P16L6: The U-Bremen MY-ice fraction is NOT "completely independent" since it uses microwave radiometer data (AMSR2 or SSMIS) at the same polarizations and frequencies as the current study.

P16L14: A RMSE -> An RMSE

P16L14: on the estimated snow depth -> between the estimated and reference snow depths

P16L15: and the snow depth -> and in-situ snow depth And you should quote the results obtained using your estimated snow depth as well since in-situ snow depths are not generally available

The discussion lacks considerations about the importance/impact of ice type.

---

## Referee Comment (RC2) · Anonymous Referee #2 · 28 Dec 2018

Review of

Estimating the snow depth, the snow-ice interface temperature, and the effective temperature of Arctic sea ice using Advanced Microwave Scanning Radiometer 2 and Ice Mass Balance buoys data

by

Kilic, L., et al.

Summary: A suite of linear regressions is derived consecutively to derive i) an estimate

of snow depth, ii) an estimate of the snow-ice interface temperature and, finally, iii) of the effective temperature Teff - all from brightness temperature (TB) observations of the AMSR2 in the Arctic Ocean during winter time. This suite is developed with the aid of TB collocated with weather forecast data, OIB snow depth data and IMB snow depth and snow-ice interface temperature observations as well as with simulations of TB, snow-ice interface temperature and Teff with a thermodynamic model in combination with a microwave emission model. Observed and retrieved snow depths and snow-ice interface temperatures are compared by means of RMSD and correlation. Examples of retrievals of the Arctic-wide distribution of snow depth and snow-ice interface temperature are shown and discussed in the context of a multiyear ice concentration product.

This paper is an interesting contribution to the scientific literature in this field. Before it could become acceptable for publication the authors need to take care of several issues which are required to understand their methodology, to potentially re-do their analysis, and to better underline the new aspects of their work in front of the background of work done by others. Solving most of these issues will help the authors to reply to my suggestions to improve their discussion of the results achieved. I therefore hand this manuscript back to the authors, asking for major revisions. The general and specific comments will potentially aid in this process.

General comments GC1: The introduction needs a better structure: Relevance - previous work - shortcomings - what will you do and why. The introduction also requires an improved set of references to make clear the current state-of-the-art of snow-depth on sea ice retrieval and also snow-ice interface temperature retrieval.

GC2: The description and illustration of the methodology to retrieve the snow depth but also in particular the snow-ice interface temperature lacks important details for the understanding. See my specific comments with this regard.

GC3: The title "promises" retrieval of effective temperature but the paper kind of stops before having applied a method retrieving it from TB data and discussing any results

into this direction.

GC4: I am a bit lost with regard to a critical discussion of the results. - I neither found a discussion about how accurate the automatically retrieved snow-ice interface temperatures are, nor did I find a discussion about the dependency of the different retrievals on the same data. For instance: The fact that in Figure 12 snow depth and multiyear ice concentration have a certain degree of correlation can partly be explained by using the same frequencies and polarizations (see eq 1 and the microwave data entering the MYI concentration maps). The same applies to Tsnow-ice, which is via its correction with the snow depth is also related to these frequencies. - Uncertainty estimates are missing in any of the retrievals presented. - A critical discussion about the physics behind the many linear regressions used would definitely add to the understanding of the paper and would give the approach more credibility.

Specific comments P2, L4-5: "Improved estimates of ... from satellite observations ..." implies that such estimates exist already. But they have not been mentioned yet.

P2, L8-19: I find this paragraph relatively weak and not suitable yet for this introduction. An improved paragraph would - more clearly separate between snow deoth and Tsnow-ice retrieval - find more references for both these parameters. Comiso et al (2003) for instance also refer to Tsnow-ice; there are other papers dealing with the application and evaluation of the Markus and Cavalieri (1998) (MandC98) approach in the Arctic; there are other papers discussing about the caveats of the MandC98 and suggesting improved retrieval, e.g. Markus et al., 2011; Kern and Ozsoy-Cicek, 2016. Isn't there a paper by Rostosky et al., 2018, also, where an alternative approach is proposed. Finally, there have been various conference contributions (Frost et al., various years) which results' possibly should not remain to be unmentioned here. All these, in my eyes, belong to the introduction. Here you motivate why you think that you approach brings added value to the research landscape in this topic. - The RRDP data set is a co-production of ESA and EU (SPICES project) activities and this needs to be mentioned. Also, to my knowledge, this data set has been published and is citable with

a doi. You might want to ask Leif Toudal Pedersen about this. - It should be mentioned that routinely processed data sets of snow depth and snow-ice interface temperature exist. It makes sense to not only check out the NSIDC data holding but also activites at JAXA and other institutions (metno for instance).

P2, L26/27: I guess it would not hurt to at least mention the substantial emissivity differences between first-year ice and multiyear ice here, i.e. the sensitivity to ice type.

P2, L31: On the one hand the "relationship ... is complicated at microwave frequencies > 18GHz" ... on the other hand "from 6 to 50 GHz [i.e. including those complicated frequencies] there is a high correlation between Teff and Tsnow-ice" This is a bit confusing and should be reformulated. Also the following statement that by "using linear regression" Teff can be estimated contradicts the previous mentioning of a complicated relationship.

P3, L12-15: - Please provide information about the product level which is used in this product and also detaiil whether swath data or gridded data are used. - For the co-location with the IMB as well as the OIB data sets it is important to know the search radius in space and time within which an IMB or OIB measurement is co-located with a satellite measurement.

P3, L15: See one of my previous comments. There should be a DOI and citable reference.

P3, L16-25: - 10 cm vertical resolution of the temperature measurements sounds a bit coarse. Please check whether there are not other (finer) vertical resolutions in different media. - How does the acoustic sounder penetrate the snow to measure the location of the snow-ice interface at 5 mm accuracy. So far I thought that these IMBs have an acoustic sounder looking downward to measure the position of the snow surface relative to the sounder and an acoustic sounder underneath the sea ice looking upwards, measuring the position of the ice underside; both together provides the total (sea ice + snow) thickness. The temperature measurements in the snow and sea ice

are then used to figure out where (approximately) the snow-ice interface is located. - How are IMB measurements co-located? What is the sampling frequency? Was there any averaging performed?

P3, L26-31: - Please provide references about the OIB data and/or the OIB campaign. - The 408 observations ... are these the 50 km sections? Do these overlap or are these consecutive sections? - Why did you use data from 2013 only? - You give precision / accuracy estimates for the IMBs but not for the OIB data. Please provide such as well for OIB.

P3, L33: "neither interpolated or smoothed ..." okay. What is the sampling in time?

P4, Table 1: - What is the mean snow depth and ice thickness given in the last two columns? Is this an average over the entire time period the buoys lived ... or over the time period from which you used the data ... or are these the initial depth and thickness values at buoy deployment? Please be more specific. - The time periods given in the second column do not last from Dec. 1 to Apr. 1. Why?

P4, L1-2: I don't understand what you want to say with this sentence. I do not rate a difference of 1-4 K over 100km as a particularly good example to state something about how variable data from adjacent buoys could be.

P5, L2-5: This is a very short description. What are the skills and limitations of this model to simulate TBs and Teff for snow-covered sea ice? Can the model deal with liquid water in the snow and/or with melt-refreeze cycles? For which microwave frequencies and polarization the model can be applied? What are the input atmospheric data? Even though the simulations were part of an earlier study it would be very helpful to have some key elements listed here.

P5, L13-19: This summary part is not very clear. - What is "forward selection"? - IMB snow depth is expressed as a function of TB using multilinear regression. ... then the IMB training data set is used to perform the regression ... ??? - "centred

(avarage was subtracted)" –> I don't understand this. Centred between what? Which average was substracted? - What TB dependence are the TSnow-ice values corrected for? - Which snow depth data set is used here? The IMB training one? - Perhaps a schematic illustration with the data flow and the different regression steps would ease understanding of your method.

P5, L25 to P6, L6: - Please explain why you use the OIB product with the much better spatial coverage and hence representation of the satellite footprint conditions only for the forward selection. Would it have been more straightforward and logical to carry out both, the forward selection AND the regression using the OIB data? What is the added value using the IMB snow depth values? - Please provide an additional table in which the results of the statistical forward selection are summarized. - Please explain the statistical measures used in the forward selection. May I ask whether you tried all frequency and polarization combinations? How many in total did you try? - Please provide at least an example, e.g. a scatterplot or 2-dimensional histogram, in which you illustrate the relationship between the 3 channels used for the best retrieval and the OIB snow depth data. It would be very intriguing to see how much the measurements scatter around the regression lines.

P7, L7-12: - It is not entirely clear what were the input and the test data sets for these additional tests of the regression. Please be more specific about what you did. This goes back to the a schematic illustration which is missing. - How does the multi-linear regression work? Is it a stepwise linear regression? If not, how do you / the method assures that with the choosen parameter combination you end up in a minimum of the multi-dimensional RMSE "surface" (optimal parameter combination)? - Is there any uncertainty involved in your parameter estimation? Or, in other words, what is the uncertainty of the SD retrieved with equation (1) based on the multilinear regression?

P6, L14-20: - "snow depth estimate from MandC98" –> Did you compute this on your own? If yes, with which coefficients? If not, where did you take the snow depth information from? Without more detailed information about this it is not possible to properly

evaluate the quality of your results. - What is the basis (in terms of time) for the intercomparison presented in this and the following paragraph? Are we talking about a 4-month average value?

P6, L21-27: - What do we know about the limitations of the MandC98 approach in terms of snow depth? - In L23/24 you kind of contradict your statement from L19/20. Please check. - I doubt that this particular buoy is located ON a ridge or hummock. In that case the local snow depth would probably not be very large because it can be expected that the wind blows the snow off the ridge and hummock. Maybe you wanted to write "nearby a ridge or hummock"? In that case your statement would be making more sense, I guess. Please check your hypothesis. - What happens with RMSE and correlation for MandC98 when skipping the data from 2013F? Please provide.

P6, L28 until P7, L2: - "uncertainties on OIB data" –> could you be a bit more specific what you mean here? How did you derive the uncertainties of the OIB data? Are these included in the product? Or are you referring to the difference between the satellite snow depth retrievals and the OIB data? - In this paragraph, as well as in the previous one and in Table 2 you are using the RMSE. In Figures 2 & 3 one can see that the difference between the IMB or OIB data on the one hand and the satellite data on the other hand can be quite large and therefore determine the RMSE. Did you try to compute an unbiased RMSE as well, by first subtracting the bias and then computing the RMSE? It might be worth a try. - The IMB data contain timeseries of snow depth which is derived from a relatively precise measurement of the location of the snow surface relative to the sounder (the downward looking acoustic sounder) and a relatively imprecise measurement of the ice-snow interface location by a temperature gradient method (to be described later in this paper apparently). Did you check the snow depth estimated from these two kinds of IMB measurements with the other data in the RRDP data set: precipitation (amount and type?) from ERA-Interim? It might be worth to do that to get a better feeling and the quality of the IMB snow depth data time series. - "Spatial scales are different" ... "the correlation is higher ..." –> yes, indeed the

scales are different. You could attempt to plot a typical satellite footprint and then, try to overplot in scale a typical OIB measurement and a typical acoustic sounder footprint. If you cannot visualize it, then it might help to quantify the difference scales again in this sentence. Another important thing which needs to be taken into account when understanding the statistics of the different data sets used is the temporal sampling which is yet not mentioned for the IMB data and which you did not specify further for the OIB data. One could argue that it is not the pure difference in spatial resolution but also and in particular the vast difference in single observations entering the one value compared between IMB, OIB and satellite data.

Table 2: - I suggest that you add the mean snow depth values as well as the standard deviation of the respective data set. The latter helps to figure out whether the data sets compared have a comparable statistics. - Am I correct assuming that the data shown in this table are only containing those IMB data which you did NOT use for the training of the method? If not that you could perhaps consider to leave these out and redo the computation. In any case it would be important to mention in the caption of the table data from which IMBs are included here.

Figure 2: - Is the length of the time series at the same scale for all IMBs? - You are also presenting the comparisong between the satellite data and the IMBs for the training data set. Is this done on purpose? - The box in the top right, annotating the figure, should be placed outside to see the full range of MandC98 snow depths. - In any case Figure 2 contains a lot information for discussion. For instance: There is not much varition in IMB snow depth for all 4 IMBs except a small step change for 2012G and a large step change for 2012H. While for 2012G both CandM98 and your approack agree with each other perfectly well, both show a clearly increasing snow depth for the other 3 IMBs, CandM98 more than your approach, an increase which in this form is not confirmed by the IMBs. - For 2013F, 2014D and 2004I your approach looks like an amplitude-dampened version of MandC98. Most of the ups and downs in the Mandc98 time series are also present in your snow depth time series. What do you think causes

Interactive
comment

the fact that the amplitude in short-terms (possibly unwanted snow depth variability) is so much smaller for your approach compared to that of MandC98? - How realistic do you think are step changes in IMB snow depth of 5 cm snow depth DECREASE as observed for 2013F and 2013G?

Figure 3: - While I doubt that an additional scatterplot with regression lines superposed does make sense for the IMB data sets I strongly recommend to add such kind of a figure here. For that it would be very good to obtain an estimate of the OIB snow depth retrieval uncertainty from the RRDP people and to estimate and uncertainty of both the MandC98 data and your approach, based on the uncertainties of the input data. Such a figure would add substantial value to the time series shown in Figure 3. - It might make sense to indicate in Figure 3 where data are over first-year ice and where over multiyear ice. - Important for Figure 2 and Figure 3 and in general all results which include MandC98 data is more information about how you used this data, i.e. whether you computed the snow depths on your own, whether you applied filters and if yes which, or whether you simply took the data out of a data base. This is important because in the products issued by NSIDC there are certain flags which, for instance flag multiyear ice because the MandC98 retrieval does not work properly there.

P7, L5 until P8, L4: - "nearly piecewise linear" sounds strange. I suggest to either write "nearly linear" or "piecewise linear". Any complicated curved profile one can approximate piecewise linear. I am not sure that this is what you wanted to express here. - "because of turbulent mixing" –> well, ok, but what if this is not existent? Then you have a strong air temperature gradient near the surface. - Is the gradient of 35K/m a typical value for the temperature gradient in snow? If so - do you have a reference? If not, then it might make sense to specify this a bit more here. Otherwise you might make the wrong assumptions in the subsequent analysis.

P8, L5-8 / Figure 4: - Only 2 of the IMBs you used show an average snow depth subtantially larger than 20-25 cm, i.e. in only two of the IMBs temperature profiles you will have more than just 2 or 3 locations where the temperature is sampled. This
does not sound a very safe method. - Figure 4 places the measurement locations exactly at the air-snow and ice-snow interfaces. I doubt that this is the case in reality. Please comment on that in the text - Figure 4 also reveals that the air-snow interface can be located quite accurately - at least in the shown setting - because the gradient change at this interface is indeed quite large. At the snow-ice interface however, the change in gradient is much smaller and almost not detectable - at least the way you plotted Figure 4. In other words, Figure 4 is not ideal to support / illustrate your method to derive the snow-ice interface from the IMB temperature profile data. See also my comment to Figure 5. - "if sea ice starts to melt" –> The isothermal state is something which is reached well after surface melt has commenced, am I right? It is hence first the temperature profile in the snow which changes before there is an isothermal state in the sea ice to be expected.

P9, L2-5: - "thermistor at the snow-ice interface" –> Please provide this detail - if confirmed by references - in the data section. It is an important detail. - "detected with our automated method" –> Did you also evaluate the success / skill of this method and can you provide a measure of its uncertainty? I'd say it is essential to know this because the high precision with which the acoustic sounder measures the location of the snow surface of 5 mm is kind of useless without knowing what the potential bias of your method to locate the ice-snow interface is. Looking at Figure 4 and the description of your method it is certainly fair to assume that a bias of 5 cm might not be uncommon.

P9, L6-12: - 89GHz TBs are highly correlated with the air-temperature –> you don't show this in any of the figures, am I right? What is this statement for? Does it add value and is it relevant for the outcome of the paper? If relevant - How well are TBs of the other frequencies correlated with the air temperature? - Yes, at 18.7 to 36.5 GHz there might be some scattering of microwave radiation in the snow. Actually it differs between 18.7 GHz and 36.5 GHz that much that it form the basis for the snow depth retrieval of the MandC98 approach. Did you know that? I would therefore - particularly because one can properly derive snow depths up to 40-50 cm depth not say

that at these frequencies one has shallow penetration into the snow. I'd rather state that for all but one of your IMB penetration at these frequencies is deep enough to properly retrieve the snow depth. I suggest to reformulate this sentence therefore to avoid contradiction and misunderstandings. - "7.3 GHz is ignored" –> but you show it in Figure 5 nevertheless. Why? - Please try to provide an explanation why the horizontally polarized channels at ∼7 and ∼11GHz have a substantially lower correlation with Tsnow-ice. - It is more than likely that at these two frequencies (7 and 11 GHz) there is also substantial penetration into the sea ice - particularly if the underlying sea ice is multiyear ice and therefore has a close to zero salinity in its uppermost centimeters to a few decameters. Actually, taking Figure 4 and 5 together suggests that what you retrieve as the ice-snow interface temperature Tsnow-ice is not necessarily exact that temperature but rather a temperature of a sea ice layer underneath - that sea ice layer into which these two low frequency channel data penetrate. –> Temperature of the effective emitting layer.

P9, L14-19: - If I understand your concept of "centred data" correctly then what you basically do is working with anomalies and compute the linear regression between the anomalies of Tsnow-ice and anomalies of the TBs at the two frequencies selected. How valid / representative is in this case your correlation analysis which you based on the absolute values and not on the anomalies. Wouldn't it have been more straightforward to carry out the correlation analysis with the data you will use at the end for your retrieval?

P9, L20/21: - I agree about the dependency of Tsnow-ice on snow depth. I do not understand, however, why you can assume that only the offset of the linear regression changes while the slope is the same for each IMB. If I take Figure 6 and draw a linear regression for each of the four IMBs used I will get different offsets AND different slopes. Please explain. - Maass et al. (2013) is a reference which certainly cites itself older references about the mentioned isolating effect of snow. Could be that the book by Untersteiner is a more appropriate reference here. - Finally: Do we expect a linear

relationship?

Equation 2: I suggest to set up this equation in the same fashion as equation 3. The way done currently is confusing. I would stick with the notation that Tsnow-ice has the form ax + b + c where ax + b are originating from the linear regressions shown in Figure 6 and c is the correction faction based on the snow depth. That you are showing the content of Eq. (2) in Figure 7 is a different thing.

P10, L5-16 / Figure 7: - When I look at Figure 7 I do not necessarily "buy" that using the inverse SD leads to an underestimation of small snow depths. I would say that the majority of data pairs of IMB 2012L fits better to the 1/SD than to the log(SD) curve. The same could be said for 2013F and the SD curve. I suggest to first remove outliers and then compute the RMSD between the fitting curve and the SD values for each IMB for each of the three fits used to have a more objective measure of the skills of the fits. These values can easily be compiled in a Table. - By the same token I recommend to discuss the physical background using these different fits. Is there perhaps evidence that one of these is particularly suitable given what we know about the interaction of microwave radiation and snow on sea ice as well as about the relationship between microwave radiation, penetration depth and Tsnow-ice? - I would highlight in the caption of Figure 7 and once more in the text that IMBs from 2012 serve as training data and that IMB data from 2013 and 2014 are independent and serve as kind of a quality check of the fits shown in Figure 7. You might even want to highlight this by choosing either different symbols or different symbol sizes. - Finally, I guess you need to explain in a bit more detail how you switch from the linear regressions given on Page 9, Lines 18/19 to equation 2 and 3 because of three (addition to my comment farther up about why only the offset changes) reasons. 1) What happens to the offset of 0.020 and 0.019? 2) The regressions obtained from Figure 6 are computed using the TB and Tsnow-ice anomalies. If I am not mistaken, you need to use the TB anomalies in Equations 2 and 3 as well then ... this is not clear. It is particularly not clear whether the Tsnow-ice value obtained with equations 3 and 4 is just the anomaly

**TCD**
[Figure]

or the "absolute" value and if the latter, where is the switch where you step back from anomaly to absolute value? 3) The snow depth you are using here ... is this the one you obtained yourself with Eq. 1 or is this (has this to be) an independent, externally provided snow depth? If it is the snow depth from Eq. 1 then at least Eq. 4 are not independent as both contain in some way information of the 6 GHz channel.

P10, L18 until P11, L7 / Figure 8 + 9: - Figure 8 and 9 only partly answer my point (3) farther up whether you need an independent (external) snow depth estimate or you can use the one retrieved with your method. - I guess it is important to discuss Figure 8 and 9 in detail. Figure 8 uses the IMB observed (or better derived) snow depth. The agreement between computed and observed (better estimated) Tsnow-ice is certainly better than in Figure 9. This needs to be stated - potentiall also in form of mean differences and standard deviations in a separate table. - I cannot see in Fig. 8 that 2012L is particularly bad. It is actually together with 2012H the IMB with the best agreement. 2012G has a positve bias (Tsnow-ice retrieved > Tsnow-ice "observed"), 2012J a negative one. 2013F and 2014F both have a negative bias while 2013G and especially 2014I have a positive bias. Is this reflected by Figure 7? - Please remain critical. Do you believe in the decrease in IMB Tsnow-ice for 2014I to -20degC until the end of the period at a mean snow depth of > 20 cm? - Is the difference between Figure 8 and 9 for 2013F and 2014I in line with the differences in retrieved snow depth versus IMB snow depth? The negative (2013F) and positive (2014I) biases become larger when going from Fig. 8 to 9. Hence the regressed snow depth has to be larger than IMB snow depth for 2013F and smaller than IMB snow depth for 2014I. Is this the case? - Since IMB snow depth estimation requires IMB Tsnow-ice, these two quantities are not independent. How useful is it then, to compare a remote sensing product which uses IMB snow depth (as a function of IMB Tsnow-ice) with the IMB Tsnow-ice itself?

P12, L3-8: - What is the ultimate goal to compare model results, which are seemingly completely independent of the observations in terms of ice type, snow depth /accumulation, and time period used (?), with your estimations of Tsnow-ice. Please provide

[Figure]

Interactive
comment

1-2 introductory sentences. Otherwise it pretty much sounds like comparing apples with oranges.

P12, L9-12: - Please use the same number of digits: 2.7 K and 2.1K instead of 2.07K. - The bias-corrected regressed Tsnow-ice values show a larger difference between 10V and 6V than found in the previous section. Why? What could be the reason? Is it because the model is capable to handle the relationship between the frequency-dependent penetration depth into the sea ice underneath the snow-ice interface and Tsnow-ice better than your estimations based on IMB-data based estimates of Tsnow-ice and its correlation with the TBs at the respective frequencies? (See my comment to figure 5).

P12, L13-15 / Figure 10: - In contrast to Figure 6 you use absolute TB and Tsnow-ice values here - while for the regressions shown in Figures 3 and 4 you (at least partly) used TB anomalies? Please explain i) why you can use the absolute values here and ii) why it is possible to use equations 3 and 4 also for the absolute values.

P13, Figure 11: Please check the caption; "at different frequencies" does not apply to the figure shown.

P13, L1/L6: "50V" ? Perhaps you write on P12, L17: "50 GHz at vertical polarization (50V)"? Then you have introduced this acronym.

P13, L3/4: I am not sure I would term this behaviour "sensitivity". It is possibly better to state - like you partly did - that if the slope of the regression is < 1 then Teff originates from below the snow-ice interface while when the regression is > 1 then Teff originates from above the snow-ice interface. This makes pretty much sense given the smaller penetration depth into snow and sea ice at 89GHz compared to the lower frequencies, i.e. 10GHz or 6 GHz.

P13, L7-9: "These linear regressions ... to retrieve the Teff ..." –> ok ... but how? Now we are at the point where I, as the reader, would like to see the "final" equation with

which I can compute Teff based on (which?) TB with (which?) external or additional input data ... Here the paper kind of stops and does not go further ahead. Why? –> GC3

P14, Table 3: Please state in the caption what the source for Tsnow-ice and Teff are.

P14, L2-8 / Figure 12 - Why do you use SIC from a weather forecast model? This is not understandable given the multitude of products available in Bremen. - You use Eq. 3 and hence first need to compute the TB anomalies ...? - Am I right in assuming that the snow-depth input into Eq. 3 is the one computed with Eq. 1 and shown in the first row of Figure 12? - You use and show a multiyear ice concentration product ... why? Is this to demonstrate / illustrate that your approach is able to compute snow depth over multiyear ice as well? While it is certainly a valuable product one gets the impression that the multiyear ice area increases during winter. Even ice drift seems not capable to explain the substantial spread of multiyear ice into the Eastern Arctic Ocean. - What is the cut-off MYI concentration value used in Figure 12, last row? In other words: What is the minimum MYI concentration displayed? It seems not to be 1%. - Please provide a measure of the actual ice cover - for instance by providing the 15% sea-ice concentration isoline in all 9 images of Figure 12. - In almost all images in Figure 12 there are tiny, noisy white dots. Where to these come from? Can you remove them?

P14, L9-13 / Figure 12 - This paragraph needs to be rewritten. I have difficulties to follow the justifications about the larger snow depth and snow depth evolution north of Greenland and the Canadian Arctic Archipelago. Yes, we know Warren et al. (1999) but there are more recent papers to check that out. Since you have been using OIB snow depth data it would be fairly easy to look into respective papers (Webster et al.) in which these data were analysed and discussed. There has also been a recent update of the Warren et al. (1999) climatology by Shalina and Sandven. Even though its data are from the past as well it is certainly worth to take a look. In addition, since the paper lacks so far the justification why - now with the new regression - also snow depth retrieval over multiyear ice is potentially possible, it would be important to get back to
this issue here and to also mention the work done by other members of the group in Bremen (Rostosky et al., Frost et al.). Nothing is specifically stated about the snow depth (quality) in the rest of the Arctic. It is in particularly not understandable why large parts of the first-year ice cover have been omitted.

P14, L14-19: - "∼-30degC" –> How do you know? Arctic wide? Which data source? - I would rethink about the November temperature you mentioned so explicitely. If it is -5 degC then the snow-ice interface temperature is colder than the atmosphere everywhere. - While it is correct that for Jan. and Apr. there are areas where a thick snow cover nicely aligns with warmer Tsnow-ice values there are also regions where a thick snow cover nicely aligns with particularly cold Tsnow-ice. This should be discussed further. - "Note that we can observe ..." –> If this is the case then this would be very confusing and I would strongly recommend to either remove or flag these areas using an appropriate sea-ice concentration threshold - appropriate in the sense that application of the flag allows a Tsnow-ice bias due to the open water of X Kelvin ... X = 2K? ..... Alternatively, you could - as has been done for the original snow depth retrieval (these people were smart) - correct the input TBs for the fraction of open water. Perhaps, by superposing 15% sea-ice concentration isolines on each image of Figure 12 helps to find out where these sea ice margins are located.

Page 15, L1 until P16, L6: - in L2: "highest values" –> of what? - in L6: "Under the same conditions, a higher ice thickness will lead to a lower Tsnow-ice value" –> really? Lets consider a 4 m thick, a 2 m thick and a 1 m thick ice flow, all at -30degC air temperature and all with 10 cm snow on top. Isn't the heat conduction through the snow the main driving factor for the ice-snow interface temperature? - In L1 next page: "positive correlation" ... I suggest to be more careful with this statement unless you can provide evidence that you indeed observe such a correlation by, e.g., picking specific subregions, compute correlations on a daily basis and present time series of these.

GC4
I give no comments to the conclusions yet as they might be rewritten after the revision.

Typos: P2, L1: "reduced" –> "reduces"

P2, L25: "of the" –> "at the"

P2, L26: "in the medium" –> perhaps better: "into the medium"?

P3 L1: "Secion" –> "Section"

P3, L9: "dataset" –> "datasets"

P14, L14 & 16: Add "C" behind the degree sign of the temperature.

P16, L5: "developped" –> "developed"

―――――――――――――――――

---

## Author Comment (AC1) · 5 Feb 2019

**Response to reviewer 1**

**We thank the reviewer (Leif Toudal Pedersen) for his helpfull comments, which improve the paper with better explanations of the methodology and important discussion about the ice types.**

General comments

[Figure]

More discussion about the impact of ice type on the results should be included. The Markus & Cavalieri snow depth algorithm is only supposed to work properly over first year ice, most of the OIB and IMB data are from areas of multi-year ice. These issues and their impact on the results should be more clearly identified and discussed.

**We have included an analysis of the ice type, and removed the comparison with Markus and Cavalieri algorithm. The IMB are located only on multiyear ice, and OIB campaigns cover first year ice and multiyear ice. We add the ice type information in Figure 3, and discuss the results in section 3.2.**

There should be a clearer wording about when the results for Tsnow-ice are derived using in-situ snow depth and when they are derived using the estimated snow-depth from this study. Both in the abstract and in the conclusions, error numbers assuming in-situ snow depth measurements are given, but these are not generally available, so the uncertainties for the retrievals using satellite snow depths are generally more relevant.

**We have remplaced the error numbers in the abstract and in the conclusion, giving the results obtained using the snow depth regression.**

The concept of effective temperature is based on an assumption of constant emissivity. It is here even referred to as surface emissivity. In reality the emissivity varies with depth as does the temperature, and in particular the emissivity at the surface is small since the emissivity of snow is very small during Winter (no absorption = no emissivity). It should be better explained what is actually the emissivity referred to as the surface emissivity, and some considerations about its variability with temperature and salinity for example would be appreciated.

**Further explanations have been added in the introduction with an equation:**

**"The surface contribution i.e., the surface brightness temperature (TB) depends on frequency and it is the product of the surface effective emissivity ($e_{eff}$) and**

the surface effective Temperature ($\mathbf{T}_{eff}$):

$$TB = e_{eff} \cdot T_{eff} \tag{1}$$

**"$\mathbf{T}_{eff}$ is defined as the integrated temperature over a layer corresponding to the penetration depth at the given frequency: the larger the wavelength, the deeper the penetration into the medium. In the same way, eeff represents the integrated emissivity over a layer corresponding to the penetration depth. It depends on the frequency, on the incidence angle, and, on the sub-surface extinction and reflections between snow and sea ice layers (Tonboe, 2010)."**

More detailed comments:

P1L20: Sea ice dynamics and thermodynamics -> Sea ice thermodynamics

**Done.**

P2L1: reduced -> reduces

**Done.**

P2L9: Advance -> Advanced

**Done.**

P2L11 and reference section: The RRDP should be referred to as Pedersen et al, 2018, https://figshare.com/articles/Reference_dataset_for_sea_ice_concentration/6626549

**Thank you, we add the reference.**

P2L24: In principle this should also be "surface effective emissivity" (see above), and it should be better explained how to estimate this emissivity.

**Better explanation has been added (see my response above).**

P3L5: See comment P2L11 above

**Reference added.**
P3L10-11: Note that neither the OIB nor the IMB data in the RRDP are guaranteed 100% ice. This should be considered and the impact on the results should be discussed.

**We verified this point. Using a SIC algorithm on Tbs (6V and 6H) at IMB position, the SIC is between 95% and 100% for all the measurements. For the OIB the SIC is also between 95% and 100% (with some lower values at 70-80%).**

P3L15: See P2L11 above. In addition the resolution matching of AMSR2 is carried out by JAXA and should be referred to as Maeda et al, 2011 Maeda, K., Y. Taniguchi and K. Imaoka, (2016), GCOM-W1 AMSR2 Level 1R Product: Dataset of Brightness Temperature Modified Using the Antenna Pattern Matching Technique, IEEE Transactions on Geoscience and Remote Sensing, VOL. 54, NO. 2.

**The references have been added.**

P3L19-20: The acoustic sounder only measures the position of the snow surface. The position of the ice surface is assumed from deployment or from the Summer measurements at the end of the ablation period. The sensor is mounted on a pole frozen into the ice, looking down at the snow surface. It measures distance between the instrument and the snow surface, thus recording the changes in the snow depth.

**On the CRREL website (http://imb-crrel-dartmouth.org/imb/), it is explained that the acoustic sounder measures the snow and the ice surface position as well as the ice bottom position. See also Richter-Menge, J. A., Perovich, D. K., Elder, B. C., Claffey, K., Rigor, I., & Ortmeyer, M. (2006). Ice mass-balance buoys: a tool for measuring and attributing changes in the thickness of the Arctic sea-ice cover. Annals of Glaciology, 44, 205-210. The reference has been added to the text.**

P3L21: IMB buoys -> IMBs. The B in IMB means Buoy and does not have to be repeated. There are many instances of this in the text.

**Ok. It has been corrected throughout the text.**

[Figure]

P3L23: bouys -> buoy

**Done.**

P3L29: OIB radar -> the OIB snow radar. OIB operates other radars as well.

**Done.**

P5L1-5: Please include a bit more details about the simulated data, such as number of datapoints, types of ice etc.

**We added more explanations:**

**"For the estimation of $T_{eff}$, we use a microwave emission model coupled with a thermodynamic model. The emission model uses the temperature, density, snow crystal and brine inclusion size, salinity, and snow or ice type to estimate the microwave emissivity, the $T_{eff}$, and the TB of sea ice. It is coupled with a thermodynamic model in order to provide realistic microphysical inputs. The thermodynamic model for snow and sea ice is forced with ECMWF ERA40 meteorological data input: surface air pressure, 2m air temperature, wind speed, incoming shortwave and longwave radiation, relative humidity, and accumulated precipitation. It computes a centimeter scale profile of the parameters used as inputs to the emission model. The emission model used here is a sea ice version of the Microwave Emission Model of Layered Snowpacks (MEMLS) (Wiesmann et al., 1999) described in Matzler et al., 2006. The simulations were part of an earlier version of the RRDP and the simulation methodology is described in Tonboe et al., 2010. This MEMLS simulation uses among its inputs the snow depth and the $T_{Snow-Ice}$ and compute $T_{effs}$ and TBs at different frequencies (from 1.4 to 183 GHz). The dataset contains 1100 cases and is called the MEMLS simulated dataset in the following."**

P5L30: satisfying -> satisfactory

**Done.**

P6L1-10: Discuss also the potential for a seasonal variation in the regression. OIB data are all from late Winter to Spring, whereas the IMB data are for all Winter. What impact could that have, and why do you expect your regression from OIB to work also during other parts of the Winter.

**The final regression for snow depth (eq 1) is computed from IMB data. The OIB data are used only for the channel selection. Therefore the regression can not be appropriate out of the winter period. We add a discussion in the results about the impact of the season on the snow depth regression for OIB data. "It is also important to note that the OIB campaign data are from late winter to beginning of spring, while IMB measurements are from winter. The snow depth regression being developed on IMB measurements, this small change in the season can contribute to the larger RMSE observed with OIB data"**

P6L21-27: Here you need to discuss why you think the Markus and Cavalieri snow depth algorithm can be applied to MY-ice.

**We know that the Markus and Cavalieri snow depth algorithm has been designed for Antarctic where the sea ice is mostly first year and young ice. The Markus and Cavalieri algorithm is based on physical and radiative properties of the snow using the 18 and 36 GHz frequencies, and we only used it to give a comparison with our algorithm. Our goal was not to evaluate the Markus and Cavalieri algorithm, so we removed it, as you suggested, because it was confusing.**

P6L31: Please provide a reference to the OIB uncertainties quoted here. Also note that the RRDP OIB dataset contains information about the variability of the snow depth over the 50 km sections. This could have been used to filter out the OIB data with too much variability. The RRDP also contains ASCAT C-band scatterometer data that could be used to distinguish ice types.

**It is the standard deviation of the OIB snow depth given in the dataset with the snow depth itself. We have added the information in the text.**

Figure 2: You should not apply the Markus and Cavalieri algorithm to MY-ice and you should discuss the importance of ice type for your own snow depth retrievals.

**We removed Markus and Cavalieri results and added the ice type information and a discussion about it. For our retrieval, the use of the 6GHz channel limits the problem of the ice type as there is not a big change in emissivity between first year and multiyear ice at this frequency.**

P7L7-8: The temperature gradient is a function of the thermal conductivities and the depth of snow and ice respectively. The temperature gradient in snow is certainly not always 35 K/m! Please rephrase this sentence.

**Yes, we removed it. That was only for one case.**

Section 4.1: This methodology is rather crude. It assumes thermodynamic equilibrium (which is not always the case, please discuss), and it could have been refined to a better estimate of the snow/ice interface temperature by the method outlined in section 4.1.5 of the RRDP manual (identifying the crossing point of the linear temperature profile in ice and in snow respectively). This might have reduced the quantization "noise" in the IMB Tsnow-ice data.

**The methodology we use is based on the same principle that the method you described in the RRDP manual. We compute the first derivative of the temperature profile to obtain the tangent then the second derivative is used to compute the variation in the temperature gradient and to identify the level in the thermistor string where the change of medium is happening. We can not use exactly the method you described as we have no a priori about which thermistor belongs to the snow and which thermistor belongs to the ice. The methodology has been designed for winter profiles and the limitations of this method are described in section 4.1.**

P10L5-14: Equations (2), (3) and (4) do not make sense as they stand. The TBs should

have been delta-TBs and you should specify the center TBs you subtracted to get to the delta TB and you should more clearly specify that these are NOT Tbs.

**These are Tbs. In the equations (2),(3),(4) we use the brightness temperature at 10V and 6V. To obtain this expression, a first step was to use the centered TB to compute the variation of the Tsnow-ice only induced by the TB. Then we use directly the TB to compute the snow depth dependence and so the final equation. We added explanations:**

**"To express the T$_{Snow-Ice}$ as a function of the TB at 6V and 10V, the linear regressions are calculated on centered data. For each buoy, the averaged T$_{Snow-Ice}$ is subtracted from the T$_{Snow-Ice}$ measurements (the same is done with the TB measurements). Thus, the temperature offset between the buoys is removed and the slope in the linear regression is unchanged.**

$$\Delta T_{Snow-Ice} = a_1 \cdot \Delta TB_{6Vor10V} \Leftrightarrow T_{Snow-Ice} = a_1 \cdot TB_{6Vor10V} + offset_{buoy} \quad (2)$$

**with $\Delta T_{Snow-Ice}$ and $\Delta TB$ describing the centered T$_{Snow-Ice}$ and TB."**

P11L5-7: This should have been mentioned earlier and could have been fixed by applying the method from the RRDP manual described above under Section 4.1.

**The problem specified here is that the vertical resolution of the thermistor string is 10cm, and the interface may not be exactly at the position of the thermistor. Even if we know exactly the position of the interface, we will need to extrapolate the temperature and this should be discuss as well.**

P12L3: Explain a bit more what Teff is and why you need simulated data.

**An explanation has been added. "Teff is related to the frequency and the incidence angle of the observations. It is not a geophysical variable that we can measure directly as an in situ parameter. A microwave emission model has to be used to computed the T ef f s from the geophysical parameters."**

P12L4: are simulated together -> are all simulated

**The sentence has been modified.**

P12L10: simulated data -> simulated TB data **Done.**

P12L13-15: These biases are presumably in the MEMLS simulations and not in the TB data, so you should bias-correct the MEMLS simulations and not the AMSR2 TB data.(This applies to figure 10)

**Here, we do not bias-correct the AMSR2 TB data. We are expressing the Tsnow-ice from MEMLS dataset as a function of the Tsnow-ice estimated from our regression (eq 3 and 4) using the TBs contained in the MEMLS dataset. We obtain an equation as follow:**

$$T_{snow-ice\ MEMLS} = Tsnow - ice - -3.97. \tag{3}$$

**Then in the following we derive the expression of Teff as a function of T$_{snow-ice\ MEMLS}$:**

$$Teff_{freq,v} = b1 \cdot T_{snow-ice\ MEMLS} + b2 \tag{4}$$

**Finally, if you want to derive the effective temperature from AMSR2 TBs you want to replace the T$_{snow-ice\ MEMLS}$ by T$_{snow-ice}$:**

$$Teff_{freq,v} = b1 \cdot (T_{snow-ice} - 3.97) + b2 \tag{5}$$

**The expressions have been added in the text to make this clearer to the reader.**

P12L20-21: Explain more (f.ex using a reference) why H pol TBs are more noisy??

**We add a explanation. Variability of the sea ice Tbs at microwave frequencies is larger in horizontal polarization that is much more sensitive to dielectric changes and to roughness (see Kilic et al. 2018).**

Figure 11: The figure text must be wrong. This figure must be for only one frequency (which)?

**It has been corrected.**

P13L4+5: As stated in the general comments, all layers emit, to the concept of "an" emitting layer is an abstraction and should be explained more carefully.

**The concept of emitting layer has been replaced by penetration depth: "A slope coefficient lower than 1 means that the penetration depth at the given frequency is deeper than snow-ice interface. At 50 GHz the slope coefficient is close to 1, meaning that the penetration depth is close to the depth of the snow-ice interface."**

P13L8: section -> sections

**Done.**

P14L10-11: According to Warren (1999) the snow depth in general is not supposed to decrease from November to January, so this reference seems wrong. If this behavior is seen in certain regions please be more specific.

**The paragraph has been re-written. "The results show that the snow depth is larger (40 cm) in the north of Greenland (Warren et al., 1999 ; Shalina and Sandven, 2018) due to the presence of drift snow caused by the numerous pressure ridges present in this area (Hanson, 1980), as anticipated. We can observe that the snow depth is larger in areas with larger multiyear ice concentrations. The variability of the snow cover is low during winter, as the snow depth reach a maximum by December and remains relatively unchanged until snowmelt (Sturm et al., 2002)."**

P16L6: The U-Bremen MY-ice fraction is NOT "completely independent" since it uses microwave radiometer data (AMSR2 or SSMIS) at the same polarizations and frequencies as the current study.

**Yes, it is the method which is independent. "an independent work done at the University of Bremen and distributed daily to users. However it should be noted**

that the input channels of both methods overlap in some AMSR2 channels, and even different channels show some covariance (Scarlat et al., 2017)."

P16L14: A RMSE -> An RMSE

**A root mean square error**

P16L14: on the estimated snow depth -> between the estimated and reference snow depths

**Done.**

P16L15: and the snow depth -> and in-situ snow depth And you should quote the results obtained using your estimated snow depth as well since in-situ snow depths are not generally available

**Yes, it has been replaced by the figures using the estimated snow depth.**

The discussion lacks considerations about the importance/impact of ice type.

**We have added a discussion about the ice types. "A RMSE of 5.1 cm is obtained between the estimated and the IMB snow depths. This snow depth retrieval is applicable for FYI and MYI, with lower uncertainties for FYI than for MYI (3.9 cm compared to 7.2 cm)." and "The errors obtained are 2.87 K and 2.90 K respectively at 10V and 6V. This T$_{Snow-Ice}$ retrieval has been tested only for MYI. It can also be applied over FYI as the 6V and 10V channels are not sensitive to the ice type (Spreen et al., 2008)."**

---

## Author Comment (AC2) · 5 Feb 2019

**Response to reviewer 2**

**We thank the reviewer for his carefull reading of the manuscript and his numerous comments which significantly improved this paper.**

Summary: A suite of linear regressions is derived consecutively to derive i) an estimate of snow depth, ii) an estimate of the snow-ice interface temperature and, finally, iii) of the effective temperature Teff - all from brightness temperature (TB) observations of

the AMSR2 in the Arctic Ocean during winter time. This suite is developed with the aid of TB collocated with weather forecast data, OIB snow depth data and IMB snow depth and snow-ice interface temperature observations as well as with simulations of TB, snow-ice interface temperature and Teff with a thermodynamic model in combination with a microwave emission model. Observed and retrieved snow depths and snow-ice interface temperatures are compared by means of RMSD and correlation. Examples of retrievals of the Arctic-wide distribution of snow depth and snow-ice interface temperature are shown and discussed in the context of a multiyear ice concentration product. This paper is an interesting contribution to the scientific literature in this field. Before it could become acceptable for publication the authors need to take care of several issues which are required to understand their methodology, to potentially re-do their analysis, and to better underline the new aspects of their work in front of the background of work done by others. Solving most of these issues will help the authors to reply to my suggestions to improve their discussion of the results achieved. I therefore hand this manuscript back to the authors, asking for major revisions. The general and specific comments will potentially aid in this process.

General comments GC1: The introduction needs a better structure: Relevance - previous work - shortcomings - what will you do and why. The introduction also requires an improved set of references to make clear the current state-of-the-art of snow-depth on sea ice retrieval and also snow-ice interface temperature retrieval.

**The introduction has been rearranged with references added following the reviewer comments. Especially, a state of the art of snow depth algorithm, and more information about the sea ice emissivity and the ice type have been added.**

GC2: The description and illustration of the methodology to retrieve the snow depth but also in particular the snow-ice interface temperature lacks important details for the understanding. See my specific comments with this regard.

**More information about the stepwise regression for the snow depth has been**

added: "We use the stepwise regression (Draper, 1998). It is a sequential pre-dictor selection technique: at each step statistic tests are computed, and the predictors included in the model are adjusted."

For Tsnow-ice, an equation with explanations has been added for the better un-derstanding of the methodology.

"Thus, the temperature offset between the buoys is removed and the slope of the linear regression is unchanged:

$$\Delta T_{Snow-Ice} = a_1 \cdot \Delta TB_{6V or 10V} \Leftrightarrow T_{Snow-Ice} = a_1 \cdot TB_{6V or 10V} + offset_{buoy} \quad (1)$$

with $\Delta T_{Snow-Ice}$ and $\Delta TB$ describing the centered **T**$_{Snow-Ice}$ and TB."

GC3: The title "promises" retrieval of effective temperature but the paper kind of stops before having applied a method retrieving it from TB data and discussing any results into this direction.

**All the equations needed to retrieve the Teff from TBs are presented in the paper and a final equation has been added to highlight it.**

"These linear regressions between the **T**$_{eff}$ and the **T**$_{Snow-Ice\ MEMLS}$ are the final step to retrieve the **T**$_{eff}$ of sea ice at microwave frequencies as a function of TBs, using the work in the previous sections to express the **T**$_{Snow-Ice}$ as a function of TBs. The biases between the AMSR2 observations and the MEMLS simulated dataset are taken into account replacing **T**$_{Snow-Ice\ MEMLS}$ by **T**$_{Snow-Ice}$ estimated from AMSR2 TBs with a bias correction (see Table 2):

$$T_{eff(freq,pol)} = b_{1(freq,pol)} \cdot (T_{Snow-Ice} - 3.97) + b_{2(freq,pol)}, \quad (2)$$

for the regression using 10V TB.

$$T_{eff(freq,pol)} = b_{1(freq,pol)} \cdot (T_{Snow-Ice} - 4.01) + b_{2(freq,pol)}, \quad (3)$$

**for the regression using 6V TB."**

GC4: I am a bit lost with regard to a critical discussion of the results. - I neither found a discussion about how accurate the automatically retrieved snow-ice interface temperatures are, nor did I find a discussion about the dependency of the different retrievals on the same data. For instance: The fact that in Figure 12 snow depth and multiyear ice concentration have a certain degree of correlation can partly be explained by using the same frequencies and polarizations (see eq 1 and the microwave data entering the MYI concentration maps). The same applies to Tsnow-ice, which is via its correction with the snow depth is also related to these frequencies. - Uncertainty estimates are missing in any of the retrievals presented. - A critical discussion about the physics behind the many linear regressions used would definitely add to the understanding of the paper and would give the approach more credibility.

**A discussion between the dependency of the retrieval of MYI and our retrievals has been added. Our retrievals have been systematically tested and compared with observations from IMB and/or OIB campaigns (see Figures 2 and 3 with section 3.2 for the snow depth retrieval and Figures 8 and 9 with section 4.4 for the snow-ice interface temperature). The errors on the snow depth and the snow ice interface temperature retrievals are mentionned also in the abstract and in the conclusion of the paper.**

Specific comments P2, L4-5: "Improved estimates of ... from satellite observations ..." implies that such estimates exist already. But they have not been mentioned yet. P2, L8-19: I find this paragraph relatively weak and not suitable yet for this introduction.An improved paragraph would - more clearly separate between snow deoth and Tsnow-ice retrieval - find more references for both these parameters. Comiso et al (2003) forinstance also refer to Tsnow-ice; there are other papers dealing with the applicationand evaluation of the Markus and Cavalieri (1998) (MandC98) approach in the Arctic;there are other papers discussing about the caveats of the MandC98 and suggesting improved retrieval, e.g. Markus et al., 2011; Kern and Ozsoy-Cicek, 2016.

Isn't there a paper by Rostosky et al., 2018, also, where an alternative approach is proposed. Finally, there have been various conference contributions (Frost et al., various years)which results possibly should not remain to be unmentioned here. All these, in my eyes, belong to the introduction. Here you motivate why you think that you approach brings added value to the research landscape in this topic. - The RRDP data set is a co-production of ESA and EU (SPICES project) activities and this needs to be mentioned. Also, to my knowledge, this data set has been published and is citable with a doi. You might want to ask Leif Toudal Pedersen about this. - It should be mentioned that routinely processed data sets of snow depth and snow-ice interface temperature exist. It makes sense to not only check out the NSIDC data holding but also activites at JAXA and other institutions (metno for instance).

**Several references have been added to better describe the algorithm state of the art. References to the RRDP has been added.**

P2, L26/27: I guess it would not hurt to at least mention the substantial emissivity differences between first-year ice and multiyear ice here, i.e. the sensitivity to ice type. Discussions about FYI and MYI have been added throughout the text. P2, L31: On the one hand the "relationship ... is complicated at microwave frequencies > 18GHz" ... on the other hand "from 6 to 50 GHz [i.e. including those complicated frequencies] there is a high correlation between Teff and Tsnow-ice" This is a bit con- fusing and should be reformulated. Also the following statement that by "using linear regression" Teff can be estimated contradicts the previous mentioning of a complicated relationship.

**It is the physical understanding which is complicated. The linear regression allows to derive the Teff at the first order.**

P3, L12-15: - Please provide information about the product level which is used in this product and also detaiil whether swath data or gridded data are used. - For the co-location with the IMB as well as the OIB data sets it is important to know the search radius in space and time within which an IMB or OIB measurement is co-located with

a satellite measurement.

**L1R AMSR2 products are used and it is swath data. For the details see the RRDP documentation.**

P3, L15: See one of my previous comments. There should be a DOI and citable reference.

**The reference and doi have been added.**

P3, L16-25: - 10 cm vertical resolution of the temperature measurements sounds a bit coarse. Please check whether there are not other (finer) vertical resolutions in different media. - How does the acoustic sounder penetrate the snow to measure the location of the snow-ice interface at 5 mm accuracy. So far I thought that these IMBs have an acoustic sounder looking downward to measure the position of the snow surface relative to the sounder and an acoustic sounder underneath the sea ice looking upwards, measuring the position of the ice underside; both together provides the total (sea ice + snow) thickness. The temperature measurements in the snow and sea ice are then used to figure out where (approximately) the snow-ice interface is located. - How are IMB measurements co-located? What is the sampling frequency? Was there any averaging performed?

**The vertical resolution of the temperature measurements is 10 cm, and there are two acoustic sounders: one above and one below the sea ice (See http://imb-crrel-dartmouth.org/imb/ and Richter-Menge, J. A., Perovich, D. K., Elder, B. C., Claffey, K., Rigor, I., & Ortmeyer, M. (2006). Ice mass-balance buoys: a tool for measuring and attributing changes in the thickness of the Arctic sea-ice cover. Annals of Glaciology, 44, 205-210.). Please see the reference to the RDDP documentation for more technical details.**

P3, L26-31: - Please provide references about the OIB data and/or the OIB campaign. - The 408 observations ... are these the 50 km sections? Do these overlap or are these

consecutive sections? - Why did you use data from 2013 only? - You give precision / accuracy estimates for the IMBs but not for the OIB data. Please provide such as well for OIB.

**Yes, the 408 observations are the 50 km section data that are provided in the RRDP. The vertical resolution of the OIB snow radar is around 3cm and the uncertainty on the snow depth is around 6 cm ( Kurtz et al., 2013).**

P3, L33: "neither interpolated or smoothed ..." okay. What is the sampling in time? The IMB provides measurements every 1-2h.

**The paragraph has been modified.**

P4, Table 1: - What is the mean snow depth and ice thickness given in the last two columns? Is this an average over the entire time period the buoys lived ... or over the time period from which you used the data ... or are these the initial depth and thicknessvalues at buoy deployment? Please be more specific. - The time periods given in thesecond column do not last from Dec. 1 to Apr. 1. Why?

**Mean snow depth and ice thickness is an average over the period specified in Table 1. This information has been added to the Table legend. The time period do not last for each buoy the Apr 1., because the buoys have been removed by the CRREL before.**

P4, L1-2: I don't understand what you want to say with this sentence. I do not rate a difference of 1-4 K over 100km as a particularly good example to state something about how variable data from adjacent buoys could be.

**The paragraph has been rewritten and shortened.**

P5, L2-5: This is a very short description. What are the skills and limitations of this model to simulate TBs and Teff for snow-covered sea ice? Can the model deal with liquid water in the snow and/or with melt-refreeze cycles? For which microwave frequencies and polarization the model can be applied? What are the input atmospheric

data? Even though the simulations were part of an earlier study it would be very helpful to have some key elements listed here.

**Informations about the model have been added:**

**"For the estimation of $T_{eff}$, we use a microwave emission model coupled with a thermodynamic model. The emission model uses the temperature, density, snow crystal and brine inclusion size, salinity, and snow or ice type to estimate the microwave emissivity, the $T_{eff}$, and the TB of sea ice. It is coupled with a thermodynamic model in order to provide realistic microphysical inputs. The thermodynamic model for snow and sea ice is forced with ECMWF ERA40 meteorological data input: surface air pressure, 2m air temperature, wind speed, incoming shortwave and longwave radiation, relative humidity, and accumulated precipitation. It computes a centimeter scale profile of the parameters used as inputs to the emission model. The emission model used here is a sea ice version of the Microwave Emission Model of Layered Snowpacks (MEMLS) (Wiesmann, 1999) described in Matzler, 2006. The simulations were part of an earlier version of the RRDP and the simulation methodology is described in Tonboe, 2010. This MEMLS simulation uses among its inputs the snow depth and the $T_{Snow-Ice}$ and compute $T_{effs}$ and TBs at different frequencies (from 1.4 to 183 GHz). The dataset contains 1100 cases and is called the MEMLS simulated dataset in the following."**

P5, L13-19: This summary part is not very clear. - What is "forward selection"? - IMB snow depth is expressed as a function of TB using multilinear regression. ... then the IMB training data set is used to perform the regression ... ??? - "centred (avarage was subtracted)" -> I don't understand this. Centred between what? Which average was substracted? - What TB dependence are the TSnow-ice values corrected for? - Which snow depth data set is used here? The IMB training one? - Perhaps a schematic illustration with the data flow and the different regression steps would ease understanding of your method.

[Figure]

**This is further explained in the respective sections of the paper and the references to the sections have been added.**

P5, L25 to P6, L6: - Please explain why you use the OIB product with the much better spatial coverage and hence representation of the satellite footprint conditions only for the forward selection. Would it have been more straightforward and logical to carry out both, the forward selection AND the regression using the OIB data? What is the added value using the IMB snow depth values? - Please provide an additional table in which the results of the statistical forward selection are summarized. - Please explain the statistical measures used in the forward selection. May I ask whether you tried all frequency and polarization combinations? How many in total did you try? - Please provide at least an example, e.g. a scatterplot or 2-dimensional histogram, in which you illustrate the relationship between the 3 channels used for the best retrieval and the OIB snow depth data. It would be very intriguing to see how much the measurements scatter around the regression lines.

**We want our snow depth algorithm to be optimized for IMB measurements. The IMBs also measure the Tsnow-ice which is one of our interest variable and the Teff is derived from the Tsnow-ice. So the OIB data were chosen for the forward selection only because the forward selection was not satisfactory with the IMB data as the snow depth variability is limited.**

**It is a stepwise forward selection. To select the most relevant AMSR2 channels, the stepwise regression (Draper,N. R., and H. Smith. Applied Regression Analysis. Hoboken. NJ: Wiley-Interscience,1998. pp. 307-312.) was used. It is a sequential parameter selection technique designed specifically for least-squares fitting. The method begins with an initial model, at each step p-value are computed and predictors included in the model are adjusted. We can constrain the number of predictors (here AMSR2 Tbs at different channels) to as many as we want.**

P6, L7-12: - It is not entirely clear what were the input and the test data sets for these additional tests of the regression. Please be more specific about what you did. This goes back to the a schematic illustration which is missing. - How does the multi-linear regression work? Is it a stepwise linear regression? If not, how do you / the method assures that with the choosen parameter combination you end up in a minimum of the multi-dimensional RMSE "surface" (optimal parameter combination)? - Is there any uncertainty involved in your parameter estimation? Or, in other words, what is the uncertainty of the SD retrieved with equation (1) based on the multilinear regression?

**It is exactly the same method as used previously (description P5 L25 to P6 L6). We use a stepwise regression to select the channels. The coefficient of the multilinear regression are then computed using a linear fit function with the least square method. The uncertainties given by the regression method itself are small compared to the error given by the comparisons with in-situ data.**

P6, L14-20: - "snow depth estimate from MandC98" -> Did you compute this on your own? If yes, with which coefficients? If not, where did you take the snow depth information from? Without more detailed information about this it is not possible to properly evaluate the quality of your results. - What is the basis (in terms of time) for the intercomparison presented in this and the following paragraph? Are we talking about a 4-month average value?

**Yes, I computed the snow depth using the AMSR2 Tbs at 19V and 37V following the equations/coefficients described in Markus and Cavalieri, 1998. The comparisons are done with the IMB data over the period given in Table 1.**

P6, L21-27: - What do we know about the limitations of the MandC98 approach in terms of snow depth? - In L23/24 you kind of contradict your statement from L19/20. Please check. - I doubt that this particular buoy is located ON a ridge or hummock. In that case the local snow depth would probably not be very large because it can be expected that the wind blows the snow off the ridge and hummock. Maybe you wanted

to write "nearby a ridge or hummock"? In that case your statement would be making more sense, I guess. Please check your hypothesis. - What happens with RMSE and correlation for MandC98 when skipping the data from 2013F? Please provide.

**Yes, it is nearby a ridge or hummock. When skipping the 2013F, the RMSE increases for the MandC98. We removed MandC98 comparison. MandC98 is not designed for Arctic. It was here as a reference for the comparison but we do not want to evaluate it.**

P6, L28 until P7, L2: - "uncertainties on OIB data" -> could you be a bit more specific what you mean here? How did you derive the uncertainties of the OIB data? Are these included in the product? Or are you referring to the difference between the satellite snow depth retrievals and the OIB data? - In this paragraph, as well as in the previous one and in Table 2 you are using the RMSE. In Figures 2 & 3 one can see that the difference between the IMB or OIB data on the one hand and the satellite data on the other hand can be quite large and therefore determine the RMSE. Did you try to compute an unbiased RMSE as well, by first subtracting the bias and then computing the RMSE? It might be worth a try. - The IMB data contain timeseries of snow depth which is derived from a relatively precise measurement of the location of the snow surface relative to the sounder (the downward looking acoustic sounder) and a relatively imprecise measurement of the ice-snow interface location by a temperature gradient method (to be described later in this paper apparently). Did you check the snow depth estimated from these two kinds of IMB measurements with the other data in the RRDP data set: precipitation (amount and type?) from ERA-Interim? It might be worth to do that to get a better feeling and the quality of the IMB snow depth data time series. - "Spatial scales are different" ... "the correlation is higher ..." -> yes, indeed the scales are different. You could attempt to plot a typical satellite footprint and then, try to overplot in scale a typical OIB measurement and a typical acoustic sounder footprint. If you cannot visualize it, then it might help to quantify the difference scales again in this sentence. Another important thing which needs to be taken into account when

understanding the statistics of the different data sets used is the temporal sampling which is yet not mentioned for the IMB data and which you did not specify further for the OIB data. One could argue that it is not the pure difference in spatial resolution but also and in particular the vast difference in single observations entering the one value compared between IMB, OIB and satellite data.

**The OIB uncertainties are included in the product and we added the reference. As we can see in Figures 2 and 3 there is not a typical bias between our regression and the IMB or OIB measurements so the results will be very close. The IMB snow depth is computed from the acoustic sounder measurements of the snow surface position and the ice surface position (and not with the snow-ice interface algorithm).**

Table 2: - I suggest that you add the mean snow depth values as well as the standard deviation of the respective data set. The latter helps to figure out whether the data sets compared have a comparable statistics. - Am I correct assuming that the data shown in this table are only containing those IMB data which you did NOT use for the training of the method? If not that you could perhaps consider to leave these out and redo the computation. In any case it would be important to mention in the caption of the table data from which IMBs are included here.

**Table 2 has been removed and the RMSEs have been computed using the IMB which are not from the training dataset.**

Figure 2: - Is the length of the time series at the same scale for all IMBs? - You are also presenting the comparisong between the satellite data and the IMBs for the training data set. Is this done on purpose? - The box in the top right, annotating the figure, should be placed outside to see the full range of MandC98 snow depths. - In any case Figure 2 contains a lot information for discussion. For instance: There is not much varition in IMB snow depth for all 4 IMBs except a small step change for 2012G and a large step change for 2012H. While for 2012G both CandM98 and your approack

agree with each other perfectly well, both show a clearly increasing snow depth for the other 3 IMBs, CandM98 more than your approach, an increase which in this form is not confirmed by the IMBs. - For 2013F, 2014D and 2004I your approach looks like an amplitude-dampened version of MandC98. Most of the ups and downs in the Mandc98 time series are also present in your snow depth time series. What do you think causes the fact that the amplitude in short-terms (possibly unwanted snow depth variability) is so much smaller for your approach compared to that of MandC98? - How realistic do you think are step changes in IMB snow depth of 5 cm snow depth DECREASE as observed for 2013F and 2013G?

**The time scale is the same for all IMBs, and it follows the description from Table 1. Our snow depth algorithm uses the 6V TB in addition to the 19V and 37V used in MandC98. The 6V TB is less sensitive to sea ice variability , than 19V and 37V Tbs. This is why our amplitude is reduced compare to MandC98. We decided to not show the MandC98 algorithm anymore in this section and to only focus on our algorithm. We also added a discussion about the ice type.**

Figure 3: - While I doubt that an additional scatterplot with regression lines superposed does make sense for the IMB data sets I strongly recommend to add such kind of a figure here. For that it would be very good to obtain an estimate of the OIB snow depth retrieval uncertainty from the RRDP people and to estimate and uncertainty of both the MandC98 data and your approach, based on the uncertainties of the input data. Such a figure would add substantial value to the time series shown in Figure 3. - It might make sense to indicate in Figure 3 where data are over first-year ice and where over multiyear ice. - Important for Figure 2 and Figure 3 and in general all results which include MandC98 data is more information about how you used this data, i.e. whether you computed the snow depths on your own, whether you applied filters and if yes which, or whether you simply took the data out of a data base. This is important because in the products issued by NSIDC there are certain flags which, for instance flag multiyear ice because the MandC98 retrieval does not work properly there.

**Yes, we added the ice type information on Figure 3 and the MandC98 algorithm has been removed.**

P7, L5 until P8, L4: - "nearly piecewise linear" sounds strange. I suggest to either write "nearly linear" or "piecewise linear". Any complicated curved profile one can approximate piecewise linear. I am not sure that this is what you wanted to express here. - "because of turbulent mixing" -> well, ok, but what if this is not existent? Then you have a strong air temperature gradient near the surface. - Is the gradient of 35K/m a typical value for the temperature gradient in snow? If so - do you have a reference? If not, then it might make sense to specify this a bit more here. Otherwise you might make the wrong assumptions in the subsequent analysis.

**Ok piecewise linear. We can work on a profile averaged over several measurements to avoid complication with the atmosphere. We suppressed the 35K/m. It was estimated just from one profile and it is not general.**

P8, L5-8 / Figure 4: - Only 2 of the IMBs you used show an average snow depth subtantially larger than 20-25 cm, i.e. in only two of the IMBs temperature profiles you will have more than just 2 or 3 locations where the temperature is sampled. This does not sound a very safe method. - Figure 4 places the measurement locations exactly at the air-snow and ice-snow interfaces. I doubt that this is the case in reality. Please comment on that in the text - Figure 4 also reveals that the air-snow interface can be located quite accurately - at least in the shown setting - because the gradient change at this interface is indeed quite large. At the snow-ice interface however, the change in gradient is much smaller and almost not detectable - at least the way you plotted Figure 4. In other words, Figure 4 is not ideal to support / illustrate your method to derive the snow-ice interface from the IMB temperature profile data. See also my comment to Figure 5. - "if sea ice starts to melt" -> The isothermal state is something which is reached well after surface melt has commenced, am I right? It is hence first the temperature profile in the snow which changes before there is an isothermal state in the sea ice to be expected.

**We know that it is not the real interface position. It is the air-snow and snow-ice interface level detected with our method as described in the Figure 4 legend. It correspond to the thermistor string level which is the closer to the snow-ice or air-snow interface. We added:**

**"The method selects the thermistor which is located the closest to the interface. Note that the real interface position can be located between two thermistors. Therefore, the shift between the real interface position and the thermistor the closest to the interface can be up to 5 cm."**

P9, L2-5: - "thermistor at the snow-ice interface" -> Please provide this detail - if confirmed by references - in the data section. It is an important detail. - "detected with our automated method" -> Did you also evaluate the success / skill of this method and can you provide a measure of its uncertainty? I'd say it is essential to know this because the high precision with which the acoustic sounder measures the location of the snow surface of 5 mm is kind of useless without knowing what the potential bias of your method to locate the ice-snow interface is. Looking at Figure 4 and the description of your method it is certainly fair to assume that a bias of 5 cm might not be uncommon.

**The acoustic sounder is used for the snow depth estimation. Here we only work on the thermistor string of IMB, and we need to know which thermistor is the closest to the interface. The vertical resolution of the IMB thermistor string is 10 cm, so the shift between the thermistor which is the closest to the interface and the real position of the interface can be of 5cm.**

P9, L6-12: - 89GHz TBs are highly correlated with the air-temperature -> you don't show this in any of the figures, am I right? What is this statement for? Does it add value and is it relevant for the outcome of the paper? If relevant - How well are TBs of the other frequencies correlated with the air temperature? - Yes, at 18.7 to 36.5 GHz there might be some scattering of microwave radiation in the snow. Actually it differs between 18.7 GHz and 36.5 GHz that much that it form the basis for the snow

depth retrieval of the MandC98 approach. Did you know that? I would therefore - particularly because one can properly derive snow depths up to 40-50 cm depth not say that at these frequencies one has shallow penetration into the snow. I'd rather state that for all but one of your IMB penetration at these frequencies is deep enough to properly retrieve the snow depth. I suggest to reformulate this sentence therefore to avoid contradiction and misunderstandings. - "7.3 GHz is ignored" -> but you show it in Figure 5 nevertheless. Why? - Please try to provide an explanation why the horizontally polarized channels at 7 and 11GHz have a substantially lower correlation with Tsnow-ice. - It is more than likely that at these two frequencies (7 and 11 GHz) there is also substantial penetration into the sea ice - particularly if the underlying sea ice is multiyear ice and therefore has a close to zero salinity in its uppermost centimeters to a few decameters. Actually, taking Figure 4 and 5 together suggests that what you retrieve as the ice-snow interface temperature Tsnow-ice is not necessarily exact that temperature but rather a temperature of a sea ice layer underneath - that sea ice layer into which these two low frequency channel data penetrate. -> Temperature of the effective emitting layer.

**Here we are looking for the most relevant channels to retrieve the Tsnow-ice. We want to explain why certain channels are not used and how we selected the 6V and 10V channels. We know that the scattering is different between 18 and 37 GHz and that there is penetration into the sea ice at 6.9 Ghz and 10.6 GHz. The point here is to choose the most relevant channels. The 7.3 GHz has been removed from Figure 5.**

P9, L14-19: - If I understand your concept of "centred data" correctly then what you basically do is working with anomalies and compute the linear regression between the anomalies of Tsnow-ice and anomalies of the TBs at the two frequencies selected. How valid / representative is in this case your correlation analysis which you based on the absolute values and not on the anomalies. Wouldn't it have been more straightforward to carry out the correlation analysis with the data you will use at the end for your

retrieval?

**Yes, these are anomalies. An equation has been added to explain this.**

$$\Delta T_{Snow-Ice} = a_1 \cdot \Delta TB_{6V or 10V} \Leftrightarrow T_{Snow-Ice} = a_1 \cdot TB_{6V or 10V} + offset_{buoy} \quad (4)$$

**with $\Delta T_{Snow-Ice}$ and $\Delta TB$ describing the centered $T_{Snow-Ice}$ and TB.**

P9, L20/21: - I agree about the dependency of Tsnow-ice on snow depth. I do not understand, however, why you can assume that only the offset of the linear regression changes while the slope is the same for each IMB. If I take Figure 6 and draw a linear regression for each of the four IMBs used I will get different offsets AND different slopes. Please explain. - Maass et al. (2013) is a reference which certainly cites itself older references about the mentioned isolating effect of snow. Could be that the book by Untersteiner is a more appropriate reference here. - Finally: Do we expect a linear relationship?

**You will certainly get different offsets and different slopes. We do not want to developp an algorithm for each buoy. We have developed a relationship which applies more generally. The references have been added.**

Equation 2: I suggest to set up this equation in the same fashion as equation 3. The way done currently is confusing. I would stick with the notation that Tsnow-ice has the form ax + b + c where ax + b are originating from the linear regressions shown in Figure 6 and c is the correction faction based on the snow depth. That you are showing the content of Eq. (2) in Figure 7 is a different thing. In equations (2),(3) and (4), the brightness temperature at 10V and 6V are used. To obtain this expression a first step was to use the centered TBs to compute the variation of the Tsnow-ice only induced by the TB. Then we use directly the TB to compute the snow depth dependence and to derive the final equation.

**We added an equation to make this clearer (see comments above).**

P10, L5-16 / Figure 7: - When I look at Figure 7 I do not necessarily "buy" that using

the inverse SD leads to an underestimation of small snow depths. I would say that the majority of data pairs of IMB 2012L fits better to the 1/SD than to the log(SD) curve. The same could be said for 2013F and the SD curve. I suggest to first remove outliers and then compute the RMSD between the fitting curve and the SD values for each IMB for each of the three fits used to have a more objective measure of the skills of the fits. These values can easily be compiled in a Table. - By the same token I recommend to discuss the physical background using these different fits. Is there perhaps evidence that one of these is particularly suitable given what we know about the interaction of microwave radiation and snow on sea ice as well as about the relationship between microwave radiation, penetration depth and Tsnow-ice? - I would highlight in the caption of Figure 7 and once more in the text that IMBs from 2012 serve as training data and that IMB data from 2013 and 2014 are independent and serve as kind of a quality check of the fits shown in Figure 7. You might even want to highlight this by choosing either different symbols or different symbol sizes. - Finally, I guess you need to explain in a bit more detail how you switch from the linear regressions given on Page 9, Lines 18/19 to equation 2 and 3 because of three (addition to my comment farther up about why only the offset changes) reasons. 1) What happens to the offset of 0.020 and 0.019? 2) The regressions obtained from Figure 6 are computed using the TB and Tsnow-ice anomalies. If I am not mistaken, you need to use the TB anomalies in Equations 2 and 3 as well then ... this is not clear. It is particularly not clear whether the Tsnow-ice value obtained with equations 3 and 4 is just the anomaly or the "absolute" value and if the latter, where is the switch where you step back from anomaly to absolute value? 3) The snow depth you are using here ... is this the one you obtained yourself with Eq. 1 or is this (has this to be) an independent, externally provided snow depth? If it is the snow depth from Eq. 1 then at least Eq. 4 are not independent as both contain in some way information of the 6 GHz channel. See also comments above for the different regression step and the switch from TB anomalies to Tbs.

**Snow and sea ice physics are complicated and we have chosen to use an empir-**

**ical model because the RRDP development made this possible. Unfortunately, the dataset is still limited. The regressions with different functions (linear, inverse, or logarithm) are very close, but we can see that the logarithmic function is the best compromise. In the future, this could be re-computed with a larger database of snow depths. The snow depth used here is the in situ snow depth provided by the IMB.**

P10, L18 until P11, L7 / Figure 8 + 9: - Figure 8 and 9 only partly answer my point (3) farther up whether you need an independent (external) snow depth estimate or you can use the one retrieved with your method. - I guess it is important to discuss Figure 8 and 9 in detail. Figure 8 uses the IMB observed (or better derived) snow depth. The agreement between computed and observed (better estimated) Tsnow-ice is certainly better than in Figure 9. This needs to be stated - potentiall also in form of mean differences and standard deviations in a separate table. - I cannot see in Fig. 8 that 2012L is particularly bad. It is actually together with 2012H the IMB with the best agreement. 2012G has a positve bias (Tsnow-ice retrieved > Tsnow-ice "observed"), 2012J a negative one. 2013F and 2014F both have a negative bias while 2013G and especially 2014I have a positive bias. Is this reflected by Figure 7? - Please remain critical. Do you believe in the decrease in IMB Tsnow-ice for 2014I to -20degC until the end of the period at a mean snow depth of > 20 cm? - Is the difference between Figure 8 and 9 for 2013F and 2014I in line with the differences in retrieved snow depth versus IMB snow depth? The negative (2013F) and positive (2014I) biases become larger when going from Fig. 8 to 9. Hence the regressed snow depth has to be larger than IMB snow depth for 2013F and smaller than IMB snow depth for 2014I. Is this the case? - Since IMB snow depth estimation requires IMB Tsnow-ice, these two quantities are not independent. How useful is it then, to compare a remote sensing product which uses IMB snow depth (as a function of IMB Tsnow-ice) with the IMB Tsnow-ice itself?

**The decrease of 20degC with 2014I IMB is measured by the thermistor of the buoy. Tsnow-ice increases with snow depth (see equations 3 and 4). The nega-**

**tive bias for 2013F is because the snow depth estimated from satellite measurements is underestimated compared to in situ measurements at the buoy location, because of local conditions. Same for the overestimation for 2012L buoy.**

P12, L3-8: - What is the ultimate goal to compare model results, which are seemingly completely independent of the observations in terms of ice type, snow depth /accumulation, and time period used (?), with your estimations of Tsnow-ice. Please provide 1-2 introductory sentences. Otherwise it pretty much sounds like comparing apples with oranges.

**We added: "The Teff is related to the frequency and the incidence angle of remote observations. It is not a geophysical variable that we can measure directly as an in situ parameter. A microwave emission model has to be used to computed the Teff from the geophysical parameters."**

P12, L9-12: - Please use the same number of digits: 2.7 K and 2.1K instead of 2.07K. - The bias-corrected regressed Tsnow-ice values show a larger difference between 10V and 6V than found in the previous section. Why? What could be the reason? Is it because the model is capable to handle the relationship between the frequency-dependent penetration depth into the sea ice underneath the snow-ice interface and Tsnow-ice better than your estimations based on IMB-data based estimates of Tsnow-ice and its correlation with the TBs at the respective frequencies? (See my comment to figure 5).

**The change has been done.**

P12, L13-15 / Figure 10: - In contrast to Figure 6 you use absolute TB and Tsnow-ice values here - while for the regressions shown in Figures 3 and 4 you (at least partly) used TB anomalies? Please explain i) why you can use the absolute values here and ii) why it is possible to use equations 3 and 4 also for the absolute values. Equations 3 and 4 use absolute Tbs values. See previous comments.

**We work with TB anomalies only for the first step of the regression. Then we work with the Tbs as it is written in equations 3 and 4. In figure 10 we plot the regression as a line by choosing a constant snow depth.**

P13, Figure 11: Please check the caption; "at different frequencies" does not apply to the figure shown.

**Yes we corrected this.**

P13, L1/L6: "50V" ? Perhaps you write on P12, L17: "50 GHz at vertical polarization (50V)"? Then you have introduced this acronym.

**Ok.**

P13, L3/4: I am not sure I would term this behaviour "sensitivity". It is possibly better to state - like you partly did - that if the slope of the regression is < 1 then Teff originates from below the snow-ice interface while when the regression is > 1 then Teff originates from above the snow-ice interface. This makes pretty much sense given the smaller penetration depth into snow and sea ice at 89GHz compared to the lower frequencies, i.e. 10GHz or 6 GHz.

**Ok the sentence has been modified.**

P13, L7-9: "These linear regressions ... to retrieve the Teff ..." -> ok ... but how? Now we are at the point where I, as the reader, would like to see the "final" equation with which I can compute Teff based on (which?) TB with (which?) external or additional input data ... Here the paper kind of stops and does not go further ahead. Why? -> GC3

**Ok we have added the final equations to retrieve Teff.**

$$T_{eff(freq,pol)} = b_{1(freq,pol)} \cdot (T_{Snow-Ice} - 3.97) + b_{2(freq,pol)}, \tag{5}$$

**for the regression using 10V TB**

$$T_{eff(freq,pol)} = b_{1(freq,pol)} \cdot (T_{Snow-Ice} - 4.01) + b_{2(freq,pol)}, \tag{6}$$

**for the regression using 6V TB**

P14, Table 3: Please state in the caption what the source for Tsnow-ice and Teff are.

**Ok it has been added. "Using Teff and Tsnow-ice provided by the simulated dataset using MEMLS and the thermodynamical model."**

P14, L2-8 / Figure 12 - Why do you use SIC from a weather forecast model? This is not understandable given the multitude of products available in Bremen. - You use Eq. 3 and hence first need to compute the TB anomalies ...? - Am I right in assuming that the snow-depth input into Eq. 3 is the one computed with Eq. 1 and shown in the first row of Figure 12? - You use and show a multiyear ice concentration product ... why? Is this to demonstrate / illustrate that your approach is able to compute snow depth over multiyear ice as well? While it is certainly a valuable product one gets the impression that the multiyear ice area increases during winter. Even ice drift seems not capable to explain the substantial spread of multiyear ice into the Eastern Arctic Ocean. - What is the cut-off MYI concentration value used in Figure 12, last row? In other words: What is the minimum MYI concentration displayed? It seems not to be 1%. - Please provide a measure of the actual ice cover - for instance by providing the 15% sea-ice concentration isoline in all 9 images of Figure 12. - In almost all images in Figure 12 there are tiny, noisy white dots. Where to these come from? Can you remove them?

**We use the SIC to filter the AMSR2 data and to consider only the areas with 100% SIC. The white dots you see are just blank because we use AMSR2 L1R swath data. The minimum MYI concentration displayed is 0% meaning that there is first year ice or no ice at all.**

P14, L9-13 / Figure 12 - This paragraph needs to be rewritten. I have difficulties to follow the justifications about the larger snow depth and snow depth evolution north of Greenland and the Canadian Arctic Archipelago. Yes, we know Warren et al. (1999) but there are more recent papers to check that out. Since you have been using OIB snow depth data it would be fairly easy to look into respective papers (Webster et al.) in

which these data were analysed and discussed. There has also been a recent update of the Warren et al. (1999) climatology by Shalina and Sandven. Even though its data are from the past as well it is certainly worth to take a look. In addition, since the paper lacks so far the justification why - now with the new regression - also snow depth retrieval over multiyear ice is potentially possible, it would be important to get back to this issue here and to also mention the work done by other members of the group in Bremen (Rostosky et al., Frost et al.). Nothing is specifically stated about the snow depth (quality) in the rest of the Arctic. It is in particularly not understandable why large parts of the first-year ice cover have been omitted.

**We added the references. The discussion about the ice type has been added (see the previous comments) The paragraph has been rewritten:**

**"The results show that the snow depth is larger (40 cm) in the north of Greenland (Warren et al., 1999; Shalina and Sandven, 2018) due to the presence of drift snow caused by the numerous pressure ridges present in this area (Hanson, 1980), as anticipated. We can observe that the snow depth is larger in areas with larger multiyear ice concentrations. The variability of the snow cover is low during winter, as the snow depth reach a maximum by December and remains relatively unchanged until snowmelt (Sturm et al., 2002)."**

**First year ice areas are not omitted, we only filter out the areas which are not 100% ice. The MYI concentration product shows the MYI concentration from 0% to 100% which mean that there are also areas with no ice at all.**

P14, L14-19: - "âĹij-30degC" -> How do you know? Arctic wide? Which data source? - I would rethink about the November temperature you mentioned so explicitly. If it is -5 degC then the snow-ice interface temperature is colder than the atmosphere everywhere. - While it is correct that for Jan. and Apr. there are areas where a thick snow cover nicely aligns with warmer Tsnow-ice values there are also regions where a thick snow cover nicely aligns with particularly cold Tsnow-ice. This should

be discussed further. - "Note that we can observe ..." -> If this is the case then this would be very confusing and I would strongly recommend to either remove or flag these areas using an appropriate sea-ice concentration threshold - appropriate in the sense that application of the flag allows a Tsnow-ice bias due to the open water of X Kelvin ... X = 2K? ..... Alternatively, you could - as has been done for the original snow depth retrieval (these people were smart) - correct the input TBs for the fraction of open water. Perhaps, by superposing 15% sea-ice concentration isolines on each image of Figure 12 helps to find out where these sea ice margins are located.

**It is ERA-interim air temperature at 2m. We added precision about the air temperature observed. We added a reference : Perovich, D. K. and Elder, B. C.: Temporal evolution of Arctic sea-ice temperature, Annals of Glaciology, 33, 207-211, 2001.**

Page 15, L1 until P16, L6: - in L2: "highest values" -> of what? - in L6: "Under the same conditions, a higher ice thickness will lead to a lower Tsnow-ice value"-> really? Lets consider a 4 m thick, a 2 m thick and a 1 m thick ice flow, all at -30degC air temperature and all with 10 cm snow on top. Isn't the heat conduction through the snow the main driving factor for the ice-snow interface temperature? - In L1 next page: "positive correlation" ... I suggest to be more careful with this statement unless you can provide evidence that you indeed observe such a correlation by, e.g., picking specific subregions, compute correlations on a daily basis and present time series of these. GC4

**"highest MYI concentrations"**

I give no comments to the conclusions yet as they might be rewritten after the revision. Typos: P2, L1: "reduced" -> "reduces" P2, L25: "of the" -> "at the" P2, L26: "in the medium" -> perhaps better: "into the medium"? P3 L1: "Secion" -> "Section" P3, L9: "dataset" -> "datasets" P14, L14 & 16: Add "C" behind the degree sign of the temperature. P16, L5: "developped" -> "developed"

**Typos corrected.**

---

## Author Response (AR3)

**Second Response to the Reviewers**

We thank the reviewers for their careful reading and their comments.

**Response to reviewer 1**

I am happy with the corrections/improvements in the paper following my suggestions/comments in the 1st round of review. However, the statement in the conclusion that "It can also be applied to FYI as the 6V and 10V channels are not sensitive to the ice type (Spreen et al., 2008)" is contradicted by Ivanova et al, 2015 (The Cryosphere, 9, 1797-1817) which show clear differences between TBs of FY and MY ice, also for 6 and 10 GHz. I suggest a less categorical wording.

**The sentence has been changed by "It can also be applied to FYI as the 6V and 10V channels have a limited sensitivity to the ice type (Comiso, 1983; Spreen et al., 2008)"**

**Response to reviewer 2**

Dear authors,

you did partly a very good job in improving the manuscript so that many passages can be understood much better now. However, I don't think it is ready for publication yet and - in my eyes - should not yet be accepted. The main reason for this is that I am still missing critical reflections on A) the inter-dependence of the used input data, on B) the uncertainty of the input data (namely the potential of OIB and IMB data being biased) and the propagation of this into the final products, and on some of the figures displaying the results (namely Figure 8, 9, and 12). In a regular paper A) and B) would be discussed in the Discussion section - which is however used to present the "end product" and a bit discussion of it. This does not replace a critical review about how reliable this suite of used regressions on partly correlated and partly potentially biased input data is. I recommend therefore, that the authors sit together for some more revisions of the manuscript before it can be accepted for publication.

Upfront I have to admit, that I am not particularly happy with the way and the degree of detail the authors have responded to my review. Many questions were ignored and hence not answered - neither in the reply to the comments nor in the paper manuscript. And a considerable amount of the questions and comments was realized in the paper manuscript in a not too convincing manner. This applies, for instance, to using literature which does not provide the basic information required (see the P2,L24 comment (see below)).

- Another example is this one: One of your replies to my review (original manuscript Page 6, Line 14-20) was: "Yes, I computed the snow depth using the AMSR2 Tbs at 19V and 37V following the equations/coefficients described in Markus and Cavalieri, 1998." and then later "We removed MandC98 comparison. MandC98 is not designed for Arctic. It was here as a reference for the comparison but we do not want to evaluate it." The way you reply to the comments is confusing. In addition, what you write is not correct. Yes, MandC98 is not designed for the Arctic. But one can of course use it over first-year ice. This has been done, there are papers about it and there are even data sets of it which are freely available. So, please also in the reply to reviewers' comments pay attention to what you write - particularly for a journal like "The Cryosphere" where the discussion can be seen by others.

**Responses to these questions have been embedded in the text. We believe it is clear now.**

- And finally this one, where you seemingly ignored a few of my questions and comments. My comments: P5, L25 to P6, L6: - Please explain why you use the OIB product with the much better spatial coverage and hence representation of the satellite footprint conditions only for the forward selection. Would it have been more straightforward and logical to carry out both, the forward selection AND the regression using the OIB data? What is the added value using the IMB snow depth values? - Please provide an additional table in which the results of the statistical forward selection are summarized. - Please explain the statistical measures used in the forward selection. May I ask whether you tried all frequency and polarization combinations? How many in total did you try? - Please provide at least an example, e.g. a scatterplot or 2-dimensional histogram, in which you illustrate the relationship between the 3 channels used for the best retrieval and the

OIB snow depth data. It would be very intriguing to see how much the measurements scatter around the regression lines.

Your reply: "We want our snow depth algorithm to be optimized for IMB measurements. The IMBs also measure the Tsnow-ice which is one of our interest variable and the Teff is derived from the Tsnow-ice. So the OIB data were chosen for the forward selection only because the forward selection was not satisfactory with the IMB data as the snow depth variability is limited. It is a stepwise forward selection. To select the most relevant AMSR2 channels, the stepwise regression (Draper,N. R., and H. Smith. Applied Regression Analysis. Hoboken. NJ: Wiley- Interscience,1998. pp. 307-312.) was used. It is a sequential parameter selection technique designed specifically for least-squares fitting. The method begins with an initial model, at each step p-value are computed and predictors included in the model are adjusted. We can constrain the number of predictors (here AMSR2 Tbs at different channels) to as many as we want."

But these are just reflections of my impression of how the authors dealt with the review. More important are the following concerns:

What the authors still fail to provide - in my eyes - is a proper discussion about the uncertainties involved. The linear regressions are taken as if they provide the truth and I have difficulties to see a critical review and discussion of the results which go into that direction. Training and evaluation with independent parts of a data set which potentially has biases does not improve the result. It only tells that the results obtained potentially have the same bias as the data used for "evaluation".

- OIB data have a certain uncertainty, are known to underrepresent thick snow over deformed sea ice and also over MYI, and to have problems with a particularly thin snow cover - as evidenced in the literature in the recent 5 years.

**Yes the uncertainties on the OIB snow depth we used in this study are given P7 L23-25 of the revised manuscript version 3. We added in the data description the reference you mention below.**

- IMB snow depth estimates have an uncertainty which is briefly mentioned in the data/methods section but which impact on the results is not further discussed.

**The precision of the snow depth measured by the acoustic sounder is 5 mm as described in the data/methods section, and it is negligeable compared to the RMSE obtained with our snow depth retrieval which is around 5 cm.**

- The interdependence of the products and methods (see my first review) by using the same channel combinations or derivates in almost all steps of the production chain is not discussed; adding one sentence I find a bit short.

**The interdependence of the products are explicit as the equations are given. It is discussed especially with the comparison of Figures 8 and 9.**

- A detailed discussion of Figures 8 and 9 is still missing - even though the authors started to give some information in the reply to my review. I want to encourage the authors to write more of this into their paper! It will give the reader the impression that the authors critically thought about the results obtained and that the authors are aware of the limitations and caveats in input data and methods. Publishing a resulting data set on a web page is, by the way, not a quality marker and cannot replace an independent quality assessment.

**The results are disccued in section 4.4.**

Other than that I have the following remaining concerns:

- Still an illustration of the suite of methods used with a diagram would strongly aid in understanding the paper. It would also illustrate much better (and perhaps solve my concern with this regard) about which parameters and input TBs enter which part of the retrieval. Such an illustration could serve as a perfect starting point to better estimate the uncertainties of the retrieved parameters which are partly depending on each other and hence errors in one parameter propagate into the next one.

**See the Figure 1 attached to this response.**

- I hoped that the concept of using a "centred" TB would have re-formulated the way it is understandable better, i.e. writing about "deviations from a mean TB" or "residual TBs" or similar.

**We added " the linear regressions are calculated on centered data (i.e. the anomaly)"**

- Once again: The produced quantities rely heavily on the usability and applicability of IMB and OIB

[Figure]

Figure 1: Flow diagram

measurements for your purpose. Therefore, even though these data are taken from the RRDP, it is in my eyes not sufficient to refer to the documentation there - particularly in case of the OIB data. This is a remote sensing product of the snow depth and not an in situ snow depth measurement.

**We added the references you suggested below.**

- Still, it is not clear what the error in the IMB snow depth measurements is and what the impact on the results is. I mean, perhaps everything is in place and readily explained but I did not find a statement like: "Deployment of the thermistor chains used in the IMBs is always such that one known thermistor is placed exactly at the snow-ice interface." *** That way the 5 mm precise measurement of the location of any surface (snow or ice) to the acoustic sounder would allow a precise measurement of the snow depth - however, at one single place only - not like with the AWI buoys deployed in the Southern Ocean where 4 snow depth measurements are made and averaged to avoid biases due to snow drift or similar - which is another error source ignored by the authors. Any statement like the one mentioned at *** would also help in the discussion of how accurate we know the snow-ice interface location and temperature.

**The IMB snow depth is measured by the acoustic sounders with 5mm precision, we added: "There are two acoustic sounders located above the snow surface and below the sea ice. The acoustic sounders measure the position of snow and ice surfaces (top and bottom) with a precision of 5 mm, from which the snow depth is computed."**

- Still, I am not particularly satisfied with the discussion of Figure 12, your main end product. One thing I have difficulties with is the strong gradient in Tsnow-ice in an area where the MYI concentration is high and where the snow depth is >= 0.4 m for both January and April cases. In addition to the >= 0.4 m snow depth areas I comment about further down in this re-review I am wondering also about those quite large areas with a snow depth below 5 cm and about the dynamics of the snow depth distribution over the course of the winter shown.

**To understand you might want to see the air temperature (see the Figure 2 below). We**

[Figure]

Figure 2: Air temperature at 2m from ERA-Interim

**modified the colorscale of Figure 12, see the response below with the histograms.**

P2, L24: Although one can find information about sea-ice emissivities in the paper by Spreen et al. (2008) I would definitely prefer citing an older reference, potentially one where these emissivities have actually been measured.

**We added a reference to Comiso, 1983. Comiso, J. C. (1983), Sea ice effective microwave emissivities from satellite passive microwave and infrared observations, J. Geophys. Res., 88(C12), 7686-7704, doi:10.1029/JC088iC12p07686.**

P3, L8-10: I am very happy with the more detailed description what Teff is. The only problem in understanding is the expression "integrated" being used for temperature as well as emissivity. What the difference between integrated and just, e.g., the mean temperature or emissivity of the respective layer?

**Temperature and emissivity can change with depth with a non linear behavior. It is this not the mean temperature or emissivity but the integration of these parameters over thickness that matters.**

P4, L8-18: In Line 12 you write "The acoustic sounder measures the position of the snow and ice surfaces with a ... computed". In the response to the reviewers' comments you clearly state that it is two sounders and one measures the location of the snow (or ice) surface and the other one measures the location of the ice underside. Both together results in the total sea ice plus snow thickness. Why is the response more detailed than what you write in the text? - The accuracy of the snow depth (and ice-snow interface location) is unknown, right? - unless one of the thermistors is placed exactly at the ice-snow interface.

**We replaced the sentence by :" There are two acoustic sounders , one is located above the snow surface and the other is located below the sea ice. They measure the position of ... computed." There are two different instrument: the acoustic sounder which measures the snow depth with a known accuracy of 5mm, and the thermistor string from which we derive the temperature of the snow-ice interface. The problem is that with the thermistor string, the thermistor chosen to measure the snow-ice interface is not necessarily located at the real snow-ice interface position, the shift can be up to 5 cm.**

Page 5, L4-5: Since the OIB data set is an important ingredient of your soup it might make sense to be very careful (and critical) with statements about resolution and uncertainty of the OIB snow depth products because in addition to (and since) Kurtz et al. (2013) quite some research has been conducted and results of that point to i) rather a double vertical resolution of the one quoted (see e.g. Kwok and Maksym, 2014) as well as ii) a minimum snow depth of about 8 cm required to be detected unambiguously (again Kwok and Makysm, 2014; Holt et al, 2015). A summary of various approaches and their limitations has been given by Kwok et al. (2017, https://doi.org/10.5194/tc-11-2571-2017). This latter reference would also be perfectly suited - particularly for OIB - for Line 10, in addition to the reference given already.

**Thank you for the references they have been added: "Recent studies evidence larger errors on OIB snow depth (Kwok and Maksym, 2014) with issues to detect snow depth under 8 cm**

(Kwok and Maksym, 2014; Holt et al., 2015). These different limitations are summarized in Kwok et al. (2017)." The exact StD of the snow depth measurements from OIB radar used in this study are given in section 3.2. "Note that the uncertainties on OIB data for the 2013 campaigns are between 2 cm and 22 cm with a mean Standard Deviation (StD) of 11 cm".

Page 8, Line 1-3: I suggest to provide the months here, i.e. March/April for OIB and October (?) to April for IMB. In addition: What you write here, lets me conclude that the "forward selection" of the relevant channels mentioned earlier, based on OIB data, is perhaps not having that much of an influence on the results.

**The months have been added.**

Page 9, Line 9-12: Yes, thank you. Exactly. And the same is true for the snow depth because a surface measurement with 5 mm precision is worth nothing (harshely put) if the location of the snow-ice interface is only known with 50 mm accuracy.

**As explained above there are two different measurements. The snow depth is known with 5 mm precision, because the acoustic sounders measure the snow surface, the top ice surface, and the bottom ice surface with 5 mm precision. The thermistor which measure the snow-ice interface temperature can be located not exactly at the real snow-ice interface and this introduces a shift of up to 5 cm with the real interface position.**

Page 12, Line 2: I doubt that Spreen et al. (2008) do make any statement about ice types in relation to 10V and 6V GHz channels of the AMSR-E instrument. Please correct and choose a correct reference. So, following your statement it doesn't matter whether I have 3 m thick MYI or 50 cm thick FYI, with a 10V or 6V GHz channel both look the same and there are no issues with different penetration depths and salinities?

**A reference to Comiso 1983 has been added. The emissivity of FYI and MYI have been computed at 6.6 and 10.7 GHz using SMMR.**

**It has been commented by Leif Toudal Pedersen also. We replaced the sentence by "...the channels 10V and 6V have a limited sensitivity to the ice type (Comiso, 1983; Spreen et al., 2008)."**

Page 15, L1 and 4/5: I would find it more logical to have the details of the MYI product mentioned directly behind its first occurrence here, i.e. the sentence in Line 4/5 should go to Line 1 after the URL of the MYI concentration data set.

**Ok. The modification has been done.**

Figure 12: Is the maximum snow depth to be retrieved 40 cm? I am asking because I am wondering about the snow depth distribution which according to the color scale the snow depth is "exactly" 40 cm. Perhaps you either shed light on this in the text or provide a histogram of the snow depth distribution for the three maps shown so that a reader can discover more details. - In addition to the white dots denoting, e.g., the meridians, there is quite some other noise in the maps of the snow depth and possibly also in the maps of the snow-ice interface temperature . Can you comment on this? Or is this simply a low-quality figure used for peer-review and the final figure will be of enhanced quality without noise / gaps?

**No it is not the maximum snow depth that can be retrieved. Looking at the snow depth histogram, the distribution is between 0 and 60 cm. The colorscale has been changed and we improved the figure quality. You can see the histograms of snow depth and Tsnow-ice in Figures 3 and 4".**

[Figure]

Figure 3: Snow depth distribution

[Figure]

Figure 4: Tsnow-ice distribution

[revised manuscript text omitted]